# Efficient Evolutionary Search Over Chemical Space with Large Language Models

**Haorui Wang**[*,1], **Marta Skreta**[*,2,3], **Cher-Tian Ser**[2], **Wenhao Gao**[4], **Lingkai Kong**[1],
**Felix Strieth-Kalthoff**[5], **Chenru Duan**[6], **Yuchen Zhuang**[1], **Yue Yu**[1], **Yanqiao Zhu**[7],
**Yuanqi Du**[†, 8], **Alán Aspuru-Guzik**[†, 2,3], **Kirill Neklyudov**[†, 9,10], **Chao Zhang**[†, 1]
[1]Georgia Institute of Technology, [2]University of Toronto, [3]Vector Institute,
[4]Massachusetts Institute of Technology, [5]University of Wuppertal, [6]Deep Principle Inc.,
[7]University of California, Los Angeles [8]Cornell University,
[9]Université de Montréal, [10]Mila - Quebec AI Institute
hwang984@gatech.edu, martaskreta@cs.toronto.edu

## Abstract

Molecular discovery, when formulated as an optimization problem, presents significant computational challenges because optimization objectives can be non-differentiable. Evolutionary Algorithms (EAs), often used to optimize black-box objectives in molecular discovery, traverse chemical space by performing random mutations and crossovers, leading to a large number of expensive objective evaluations. In this work, we ameliorate this shortcoming by incorporating chemistry-aware Large Language Models (LLMs) into EAs. Namely, we redesign crossover and mutation operations in EAs using LLMs trained on large corpora of chemical information. We perform extensive empirical studies on both commercial and open-source models on multiple tasks involving property optimization, molecular rediscovery, and structure-based drug design, demonstrating that the joint usage of LLMs with EAs yields superior performance over all baseline models across single- and multi-objective settings. We demonstrate that our algorithm improves both the quality of the final solution and convergence speed, thereby reducing the number of required objective evaluations.

## 1 Introduction

Molecular discovery is a complex and iterative process involving the design, synthesis, evaluation, and refinement of molecule candidates. This process is often slow and laborious, making it difficult to meet the increasing demand for new molecules in domains such as pharmaceuticals, optoelectronics, and energy storage (Tom et al., 2024). One significant challenge is that evaluating molecular properties often requires expensive evaluations (oracles), such as wet-lab experiments, bioassays, and computational simulations (Gensch et al., 2022; Stokes et al., 2020). Even approximate computational evaluations require substantial resources (Gensch et al., 2022). Consequently, the development of efficient algorithms for molecular search, prediction, and generation has gained traction in chemistry to accelerate the discovery process. These advancements in computational techniques, particularly machine learning-driven methods, have facilitated the rapid identification and proposal of promising molecular candidates for real-world experiments (Kristiadi et al., 2024; Atz et al., 2021; Du et al., 2024; Nigam et al., 2023).

Several current approaches used to generate molecular candidates are based on Evolutionary Algorithms (EAs) (Holland, 1992), which do not require the evaluation of gradients and are thus well-suited for black-box objectives in molecular discovery. However, a major downside is that they generate proposals randomly without leveraging task-specific information. Consequently, producing reasonable candidates requires numerous evaluations of the objective function, limiting the practical application of these algorithms. Thus, proposals generated by operators that incorporate task-specific information can help reduce the number of evaluations required to optimize the objective function.

---

[*]Equal first-authorship, [†] Equal senior-authorship

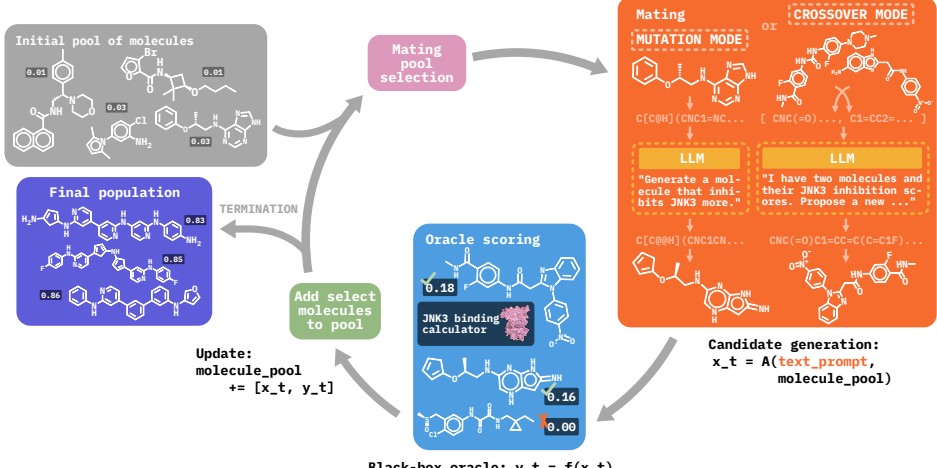

Figure 1: Overview of MOLLEO. Given an initial pool of molecules, mates are selected using default Graph-GA (Jensen, 2019) heuristics and converted to SMILES or SELFIES strings. LLMs then function as mutation or crossover operators, editing the molecule string representations based on text prompts that describe the target objective(s). The offspring molecules are then evaluated using an oracle, and the best-scoring ones are passed to the next generation. This process is repeated until the maximum number of allowed molecule evaluations is performed.

Natural language processing (NLP) has increasingly been utilized to represent molecular structures (Chithrananda et al.; Schwaller et al., 2019; Öztürk et al., 2020) and extract chemical knowledge from literature Tshitoyan et al. (2019). The connection between NLP and molecular systems is facilitated by molecular representations such as the Simplified Molecular Input Line Entry System (SMILES) and Self-Referencing Embedded Strings (SELFIES) (Weininger, 1988; Daylight Chemical Information Systems, 2007; Krenn et al., 2020). These methods convert 2D molecular graphs into text, allowing molecular structures to be represented in the same modality as their textual descriptions.

Recently, the performance of Large Language Models (LLMs) has been investigated in several chemistry-related tasks, such as predicting molecular properties (Guo et al., 2023b; Jablonka et al., 2024), retrieving optimal molecules (Kristiadi et al., 2024; Ramos et al., 2023; Ye et al., 2023), automating chemistry experiments Bran et al. (2023); Boiko et al. (2023); Yoshikawa et al. (2023); Darvish et al. (2024), and generating molecules with target properties (Flam-Shepherd & Aspuru-Guzik, 2023; Liu et al., 2024; Ye et al., 2023). Because LLMs have been trained on large corpora of text that include a wide range of tasks, they demonstrate general-purpose language comprehension as well as knowledge of basic chemistry, making them interesting tools for chemical discovery tasks (White, 2023). However, many LLM-based approaches depend on in-context learning and prompt engineering (Guo et al., 2023b). This can pose issues when designing molecules with strict numerical objectives, as LLMs may struggle to satisfy precise numerical constraints or optimize for specific numerical targets (AI4Science & Quantum, 2023). Furthermore, methods that solely depend on LLM prompting may produce molecules with lower fitness due to a lack of physical grounding, or they may produce invalid SMILES that cannot be decoded into chemical structures (Skinnider, 2024).

In this work, we propose **Mol**ecular **L**anguage-Enhanced **E**volutionary **O**ptimization (MOLLEO), which incorporates LLMs into EAs to enhance the quality of generated proposals and accelerate the optimization process (see Figure 1). MOLLEO leverages LLMs as genetic operators to produce new proposals through crossover or mutation. To our knowledge, this is the first demonstration of how LLMs can be incorporated into EA frameworks for molecular generation. In this work, we consider three LLMs: GPT-4 (Achiam et al., 2023), BioT5 (Pei et al., 2023), and MoleculeSTM (MolSTM) (Liu et al., 2023b). We integrate each LLM into separate crossover and mutation procedures, justifying our design choices through ablation studies. We empirically demonstrate the superior performance of MOLLEO across multiple black-box optimization tasks, including single-objective and multi-objective optimization. For all tasks, including more challenging ones like protein-ligand docking, MOLLEO outperforms the baseline EA and other optimization algorithms based on reinforcement learning (RL) and Bayesian Optimization (BO). To further illustrate how our model can be used in novel molecular discovery settings, we show that MOLLEO can improve on the best existing JNK3 inhibitor molecules in ZINC 250K (Sterling & Irwin, 2015).

## 2 RELATED WORK

### 2.1 MOLECULAR OPTIMIZATION

The molecular design field, encompassing multiple fundamental problems in chemistry, has developed numerous methods. In general, all the existing approaches define the space of possible molecular structures and run a combinatorial search to find the molecule with the target properties. Namely, conventional methods include Monte Carlo Tree Search (MCTS) (Yang et al., 2017), Reinforcement Learning (RL) (Olivecrona et al., 2017a; Guo & Schwaller, 2023), and Genetic Algorithms (GA) (Jensen, 2019; Fu et al., 2021; Nigam et al., 2022; Fu et al., 2022).

Due to existing challenges such as searching through exponentially large chemical space and evaluating expensive objectives (Bohacek et al., 1996; Stumpfe & Bajorath, 2012), conventional algorithms have recently recoursed to machine learning techniques, especially generative modeling (Du et al., 2024). Generative models learn a probability distribution of the observed data which can be later used to propose new molecular structures, thereby concentrating the search space around valid molecular structures. Depending on the type of the data and necessary properties for the search algorithms, different generative models have been considered: autoregressive models (ARs) (Popova et al., 2019; Gao et al., 2021), variational autoencoders (VAEs) (Gómez-Bombarelli et al., 2018; Jin et al., 2018), flow-based models Madhawa et al. (2019); Shi et al. (2020), diffusion models Hoogeboom et al. (2022); Schneuing et al. (2022).

Despite concentrating the search space around valid molecules by the usage of generative modeling, the optimization of necessary properties can remain infeasible. To narrow down the search space further, one can consider the conditional generative modeling, where the molecular structures are sampled from the conditional distribution having some predefined properties (Gómez-Bombarelli et al., 2018; Griffiths & Hernández-Lobato, 2020; Zang & Wang, 2020; Du et al., 2022; Wei et al., 2024). In this paper, we demonstrate the use of chemistry-aware LLMs as conditional generative models that improve the efficiency of combinatorial search in the molecular space.

### 2.2 LANGUAGE MODELS IN CHEMISTRY

LLMs have been widely investigated for their applicability in scientific domains (Achiam et al., 2023; AI4Science & Quantum, 2023), as well as their ability to leverage chemistry tools for chemical discovery and characterization (Bran et al., 2023; Boiko et al., 2023). Several works have benchmarked LLMs such as GPT-4 on chemistry tasks and found that while LLMs can outperform human chemists in some zero-shot question-answering settings, they still struggle with chemical reasoning (Mirza et al., 2024; Guo et al., 2023b). Several smaller, open-source models have been trained or fine-tuned specifically on chemistry text (Taylor et al., 2022; Christofidellis et al., 2023; Pei et al., 2023).

Recently, language models have also been used to guide a given input molecular structure towards specific objective properties; a widely-used term used for this is *molecular editing* (Liu et al., 2023b; Ye et al., 2023). Modifying structures towards specified properties is important so that they can satisfy potentially many required criteria, a requirement in pharmaceutical development where molecules need to be non-toxic and effective against their target (among other things), or in battery design, where molecules need to have a large energy capacity and a long lifespan. In this paper, we focus on molecular optimization to find molecules with desired properties, rather than editing. For interested readers, we provide additional related works about how LLMs have been combined with EAs for code and text generation, as well as benchmarking LLMs in chemical tasks in Appendix A.1.

## 3 THE MOLLEO FRAMEWORK

### 3.1 PROBLEM STATEMENT

**Black-box optimization.** Molecule discovery with a given property can be formulated as an optimization problem

$$m^* = \arg \max_{m \in M} F(m) \tag{1}$$

where $m$ is a molecular structure and $M$ denotes the set of valid molecules constituting the entire chemical space. The objective $F(m) : M \to \mathbb{R}$ is a black-box scalar-valued function that measures a certain molecule property $m$.

The measurement of chemical properties can involve complicated simulations or *in vivo* experiments, making it impossible to evaluate the gradients of the objective function $F$. Additionally, we assume that the main computational expense of the optimization procedure comes from the objective evaluation (oracle call). Therefore, we design algorithms to minimize the number of oracle calls and compare all the algorithms with the same call budget.

**Multi-objective black-box optimization.** Oftentimes, molecules need to meet multiple, potentially competing objectives simultaneously. Multi-objective optimization aims to find the Pareto-optimal solution, where none of the objectives can be improved without deteriorating any of them (Lin et al., 2022). The naive approach to optimize given objectives $\{F_i(\cdot)\}_{i=1}^n$ jointly is to consider an aggregate objective, such as the sum of all individual objectives, i.e.

$$m^* = \arg \max_{m \in M} \sum_i w_i F_i(m) \,, \tag{2}$$

where $w_i$ is the weight of $i$-th objective, which can be considered a hyperparameter. However, determining the weight of each objective function might be nontrivial (Kusanda et al., 2022).

The rigorous approach to multi-objective optimization is the introduction of partial order and considering the solutions from the Pareto frontier (Geoffrion, 1968; Ekins et al., 2010). In this context, the partial order is defined by comparing all the objectives $\{F_i(\cdot)\}_{i=1}^n$ for the given molecules, i.e., $m'$ surpasses $m$ if every objective evaluated on $m'$ is greater than the same objective evaluated on $m$ (assuming the maximization of objectives). Formally,

$$m' \succeq m \iff \forall i \ \ F_i(m') \geq F_i(m) \,. \tag{3}$$

For the given set of molecules $S = \{m_j\}_{j=1}^m$, the Pareto frontier $P(S)$ is defined as the set of non-dominated solutions. Namely, for every molecule $m \in P(s)$ there is no other molecule in $S$ surpassing $m$, i.e.

$$P(S) = \{m \in S : \{m' \in S : m' \succeq m \,, \ m' \neq m\} = \varnothing\} \,. \tag{4}$$

When jointly optimizing several objectives, we use the Pareto frontier to select candidates during the evolutionary search and compare algorithms. Namely, assuming that the objectives are bounded (e.g., $F(\cdot) \in [0, 1]$), one can compare two Pareto frontiers by evaluating their hypervolume

$$\text{Volume}(P(S)) = \text{Volume}\left(\cup_{m \in P(s)} H(m)\right) \,, \quad H(m) = \{x \in [0, 1]^n : x_i \leq F_i(m) \,, \forall i\} \,, \tag{5}$$

where $H(m)$ is the hyperrectangle associated with the objectives evaluated on molecule $m$, and $\text{Volume}(\cdot)$ evaluates the Euclidean volume of the input set.

## 3.2 EVOLUTIONARY ALGORITHMS

We build our MOLLEO framework upon the Graph-GA algorithm (Jensen, 2019) — an evolutionary algorithm that operates as follows. An initial pool of molecules is randomly selected, and their fitnesses are calculated using a black-box oracle, $F(\cdot)$. Two parents are then sampled with a probability proportional to their fitnesses and combined using a CROSSOVER operator to generate an offspring, followed by a random MUTATION with probability $p_m$. This process is repeated num_crossover times, and the children are added to the pool of offspring. Finally, the fitnesses of the offspring are measured using $F(\cdot)$ and the offspring are added to the population. For single-objective optimization, the $n_c$ fittest members from the population at a given step are selected to pass on to the next generation. For multi-objective optimization, two strategies are investigated: (1) Objective summation, where the summation of individual objectives is used as a single objective, and the $n_c$ fittest members are retained; and (2) Pareto set selection, where only the Pareto frontier of the current population is kept. This process is repeated until the maximum allowed oracle calls (oracle budget) have been made. This process is outlined in Algorithm 1.

We incorporate chemistry-aware LLMs into the structure of Graph-GA by using them as proposal generators at CROSSOVER and MUTATION steps. That is, for the CROSSOVER step, instead of randomly

---

**Algorithm 1:** MOLLEO Algorithm

---

**Data:** the initial pool $\mathbb{M}_0$; the objective $F$; the population size $n_c$; the number of offspring $n_o$.
**Result:** Optimized molecule population $\mathbb{M}^*$
**begin**
    **for** $m \in \mathbb{M}_0$ **do**
        Compute $F(m)$;
    **for** $t \in [1, \texttt{oracle\_budget}]$ **do**
        offspring = [];
        **for** num_crossovers **do**
            sample $m_0, m_1$ from $\mathbb{M}_t$ proportionally to objective value $F(m)$;
            offspring.append(CROSSOVER($m_0, m_1$));
        $\mathbb{M}_t \leftarrow$ sorted($\mathbb{M}_t$);
        **for** $i \in [1, \texttt{num\_mutations}]$ **do**
            offspring.append(MUTATION($\mathbb{M}_t[i]$));
        offspring $\leftarrow$ search(offspring)[: $n_o$]        ▷ smallest Tanimoto distance to $\mathbb{M}_t[0]$
        $\mathbb{M}_t \leftarrow$ offspring;
        **for** $m \in \mathbb{M}_t$ **do**
            Compute $F(m)$;
        **if** Task_type == single_objective **then**
            $\mathbb{M}_t \leftarrow$ sorted($\mathbb{M}_t$)[: $n_c$];
        **else**
            $\mathbb{M}_t \leftarrow$ Pareto_Frontier($\mathbb{M}_t$);
    Return $\mathbb{M}_t$;

---

combining two parent molecules, we generate molecules that maximize the objective fitness function guided by the objective description. For the MUTATION step, the operator mutates the fittest members of the current population based on the target description. However, we noticed that LLMs do not always generate candidates with higher fitness than the input molecule (demonstrated in Appendix C.1), and so we constructed a selection pressure to filter edited molecules based on structural similarity to the top molecule (Nigam et al., 2022). That is, we sort the existing population by fitness, apply a mutation to the top population members, and then add them to the pool of offspring. Then, we prune the pool by selecting the $n_o$ most similar offspring to the fittest molecule in the entire pool based on Tanimoto distance. We ablate the impact of this filter in Appendix C.2.

For each LLM, we describe below the details of how we implement the CROSSOVER and MUTATION operators. We empirically studied different combinations of models and hyperparameters (Appendix C.2), and in what follows, we describe the operators that resulted in the best performance.

**Graph-GA** The baseline algorithm that we build upon and compare against in our experiments.

▷ CROSSOVER: (default Graph-GA crossover): Two parent molecules are sampled with a probability proportional to their fitness. Crossover takes place at a ring or non-ring position with equal likelihood. Parents are cut at random positions into fragments, and then fragments from both parents are combined. Invalid molecules are filtered out, and a randomly spliced molecule is returned Jensen (2019).
▷ MUTATION: (default Graph-GA mutation): Random operations such as bond insertion or deletion, atom insertion or deletion, bond order swapping, or atom identity changes are done with predetermined likelihoods Jensen (2019).

**MOLLEO (GPT-4)** GPT-4 is a proprietary LLM trained on a web-scale text corpus.

▷ CROSSOVER: Two parent molecules are sampled the same way as in Graph-GA. GPT-4 is then prompted to generate an offspring with the template $t_{in}$ = "I have two molecules and their [target_objective] scores: ($s_{in,0}$, $f_0$), ($s_{in,1}$, $f_1$). Propose a new molecule with a higher [target_objective] by making crossover and mutations based on the given molecules." , where $s_{in,x}$ is an input SMILES and $f_x$ is its fitness score. This prompt template is similar to those found in AI4Science & Quantum (2023); all prompts can be found in Appendix E. We then obtain an edited SMILES molecule as an output: $s_{out}$ = GPT-4($t_{in}$). If $s_{out}$ cannot be

decoded to a valid molecule structure, we generate an offspring using the default crossover operation from Graph-GA. We demonstrate the frequency of invalid LLM edits in Appendix C.1.

▷ MUTATION: While GPT-4 performs well as a MUTATION operator when paired with GPT-4 CROSSOVER (Appendix C.2), we found that the default Graph-GA mutation achieves comparable performance with fewer LLM queries. Therefore, we opt to use the default Graph-GA mutation here.

**MOLLEO (BIOT5)** BioT5 was developed with a two-phase training process using a baseline T5 model (Raffel et al., 2020). Initially, the model was trained on molecule-text data (339K samples), SELFIES structures, protein sequences, and general scientific text from multiple sources (Pei et al., 2023) using language masking as a training objective. Following this, the model was fine-tuned on specific downstream tasks, including text-based molecular generation, where molecules are generated given an input description (Edwards et al., 2022).

▷ CROSSOVER: We use the default Graph-GA crossover.

▷ MUTATION: For the top $Y$ molecules in the entire pool, we mutate them by prompting BioT5 with the template $t_{in}$ = "Definition: You are given a molecule SELFIES. Your job is to generate a SELFIES molecule that [target_objective]. Now complete the following example – Input: <bom>[$l_{in}$]<eom> Output", where $l_{in}$ is the SELFIES representation of a molecule. These prompts have the same format as those proposed in Pei et al. (2023); exact prompts for all tasks are in Appendix E. We then obtain an edited SELFIES molecule as an output: $l_{out}$ = BioT5($t_{in}$). We transform $l_{out}$ back to the SMILES representation and add it to the pool of offspring. Since SELFIES can always be decoded into a molecular structure, there are no issues with BioT5 generating invalid molecules. With $X$ offspring produced from crossover and $Y$ offspring from the editing procedure, we select the top $n_c$ offspring overall. This selection is based on structural similarity determined using Tanimoto distance to the fittest molecule in the entire pool Nigam et al. (2022).

**MOLLEO (MOLSTM)** MoleculeSTM was developed by jointly training molecule and text encoders on molecule-text pairs from PubChem using a contrastive loss, which maximizes the embedding similarity of each pair (Liu et al., 2023b). To enable molecular editing, they implemented a simple adaptor module to align their molecule encoder with the encoder of a pre-trained generative model. This alignment allowed them to utilize the generative model's decoder for structure generation.

▷ CROSSOVER: We use the default Graph-GA crossover.

▷ MUTATION: For the top $Y$ molecules in the entire pool, we edited them by following a single text-conditioned editing step from (Liu et al., 2023b). Given the MoleculeSTM molecule and text encoders ($E_{Mc}$ and $E_{Tc}$, respectively), a pre-trained generative model consisting of an encoder $E_{Mg}$ and decoder $D_{Mg}$ (Irwin et al., 2022), and an adaptor module ($A_{gc}$) to align embeddings from $E_{Mc}$ and $E_{Mg}$, an input molecule SMILES ($s_{in}$) is edited towards a text prompt describing the objective by updating the embedding from $E_{Mg}$. First, the molecule embedding $x_0$ is obtained from $E_{Mg}(s_{in})$. Then, $x_0$ is updated using gradient descent for $T$ iterations:

$$x_{t+1} = x_t - \alpha \nabla_{x_t} \mathcal{L}(x_t),  \tag{6}$$

where $\alpha$ is the learning rate and $\mathcal{L}(x_t)$ is defined as:

$$\mathcal{L}(x_t) = -\texttt{cosine\_sim}\left(E_{Mc}(A_{gc}(x_t)), E_{Tc}(\texttt{text\_prompt})\right) + \lambda ||x_t - x_0||_2.  \tag{7}$$

$\lambda$ controls how much the embedding at iteration $t$ can deviate from the input embedding. Finally, $x_T$ is passed to the decoder $D_{Mg}$ to generate a molecule SMILES $s_{out}$. The text prompts follow the format "This molecule {has/is/other verb property}", which follows Liu et al. (2023b); exact prompts for all tasks are in Appendix E. We ablate MolSTM hyperparameter selection in Appendix C.4. If $s_{out}$ cannot be decoded into a valid molecule (see Appendix C.1), we edit the next best molecule (so that we have $Y$ offspring after the editing has finished). Similarly to MOLLEO (BIOT5), we combine the $X$ crossover and $Y$ mutated offspring and select the $n_c$ most similar molecules to the top molecule overall to keep.

## 4 EXPERIMENTS

### 4.1 EXPERIMENTAL SETUP

We evaluate MOLLEO on 26 tasks from two molecular generation benchmarks, Practical Molecular Optimization (PMO) (Gao et al., 2022) and Therapeutics Data Commons (TDC) (Huang et al., 2021). Exact task definitions can be found in TDC [1]. We organize the tasks into the following categories:

1. *Structure-based optimization*, which optimizes for molecules based on target structures. It includes isomer generation based on a target molecular formula (`isomers_c7h8n2o2`, `isomers_c9h10n2o2pf2cl`) and two tasks based on matching or avoiding scaffolds and substructure motifs (`deco_hop`, `scaffold_hop`, `valsartan_smarts`).

2. *Name-based optimization*. These tasks involve finding compounds similar to known drugs (Mestranol, Albuterol, Thiothixene, Celecoxib, troglitazone) and seven multi-property optimization tasks (MPO) that aim to rediscover drugs (Perindopril, Ranolazine, Sitagliptin, Amlodipine, Fexofenadine, Osimertinib, Zaleplon) while optimizing for other properties such as hydrophobicity (LogP) and permeability (TPSA). Two tasks, `median1` and `median2`, aim to generate molecules with properties similar to several known drugs simultaneously. Successfully completing these tasks means that LLMs can make perturbations toward desired molecules when given a chemical optimization goal.

3. *Property optimization*. We first consider the trivial property optimization task `QED` (Bickerton et al., 2012), which measures the drug-likeness of a molecule based on a set of simple heuristics. We then focus on the three tasks that measure a molecule's activity against the following proteins: DRD2 (Dopamine receptor D2), GSK3$\beta$ (Glycogen synthase kinase-3 beta), and JNK3 (c-Jun N-terminal kinase-3). For these tasks, molecular inhibition is determined by pre-trained classifiers that take in a SMILES string and output a value $p \in [0, 1]$, where $p \geq 0.5$ predicts that a molecule inhibits protein activity. Finally, we include three protein-ligand docking tasks from TDC (Graff et al., 2021) (also referred to as structure-based drug design (Kuntz, 1992)), which are more difficult tasks closer to real-world drug design compared to simple physicochemical properties (Cieplinski et al., 2020). The proteins we consider are DRD3 (dopamine receptor D3, PDB ID: 3PBL), EGFR (epidermal growth factor receptor, PDB ID: 2RGP), and Adenosine A2A receptor (PDB ID: 3EML). Molecules are docked against the protein using AutoDock Vina (Eberhardt et al., 2021), with the output being the docking score of the binding process.

To evaluate our method, we follow (Gao et al., 2022) and report the area under the curve of top-$k$ average property values versus the number of oracle calls (AUC top-$k$), which takes into account both the objective values and the computational budget spent. For this study, we set $k = 10$ in order to identify a small, distinct set of top molecular candidates. For the multi-objective optimization, we consider two metrics: top-10 AUC for summing all optimized objectives and the hypervolume of the Pareto frontier (see Equation (5)).

For baselines, we use the highest three ranking models from the PMO benchmark (Gao et al., 2022), including REINVENT (Olivecrona et al., 2017b), an RNN that utilizes a reinforcement learning-based policy to guide generation; Graph-GA; and Gaussian process Bayesian optimization (GP BO) (Tripp et al., 2021), where a GP acquisition function is optimized with methods from Graph-GA. We also include Augmented Memory (Guo & Schwaller, 2024), which combines data augmentation with experience replay to enhance the reinforcement learning-based policy for guiding generation, as well as Differentiable Scaffolding Tree for Molecule Optimization (DST, (Fu et al., 2021)), which optimizes a molecule structure using gradient ascent in the latent space of a graph neural network trained to predict a target property.

For the initial population of molecules, we randomly sample 120 molecules from ZINC 250K (Sterling & Irwin, 2015). In all runs, we restrict the budget of oracle calls to $10,000$ but terminate the algorithm early if the average fitness of the top-100 molecules does not increase by $10^{-3}$ within 5 epochs, as was done in (Gao et al., 2022). For the docking experiments, we restrict the budget to 1000 calls due to higher evaluation costs. Additional experimental details and the choice of hyperparameters are provided in Appendix B.

---

[1]https://github.com/mims-harvard/TDC/blob/main/tdc/chem_utils/oracle/oracle.py

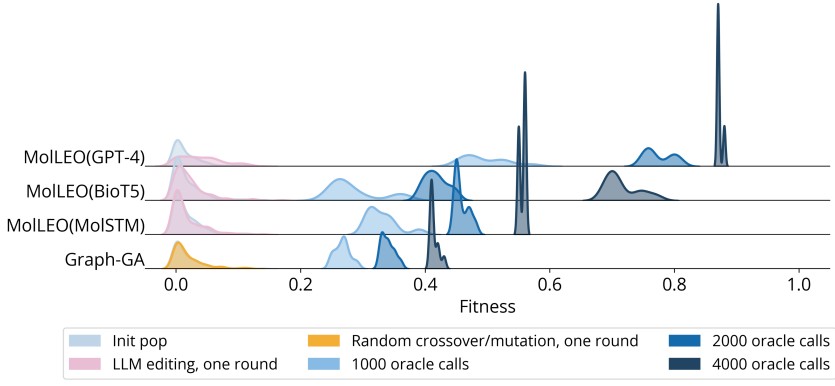

Figure 2: Population fitness over increasing number of iterations for JNK3 inhibition. In the lightest blue, we plot the fitness distribution of the initial molecule pool. We then pass the molecules through a single round of LLM edits (pink curve), or a single round of random crossover/mutation operations (yellow curve). We then show the fitnesses of the top-10 molecules after 1000-4000 oracle calls.

Table 1: Top-10 AUC of single-objective tasks. The best model for each task is bolded and the top three are underlined. We also report the sum of all tasks (total) and the rank of each model overall.

| Task type | Method objective (↑) | REINVENT | Augmented Memory | Graph GA | GP BO | MOLLEO (MolSTM) | MOLLEO (BioT5) | MOLLEO (GPT-4) |
|---|---|---|---|---|---|---|---|---|
| Property optimization | QED | 0.941 ± 0.000 | 0.941 ± 0.000 | 0.940 ± 0.000 | 0.937 ± 0.000 | 0.937 ± 0.002 | 0.937 ± 0.002 | **0.948 ± 0.000** |
| | JNK3 | 0.783 ± 0.023 | 0.773 ± 0.073 | 0.553 ± 0.136 | 0.564 ± 0.155 | 0.643 ± 0.226 | 0.728 ± 0.079 | **0.790 ± 0.027** |
| | DRD2 | 0.945 ± 0.007 | 0.962 ± 0.005 | 0.964 ± 0.012 | 0.923 ± 0.017 | 0.975 ± 0.003 | **0.981 ± 0.002** | 0.968 ± 0.012 |
| | GSK3β | 0.865 ± 0.043 | 0.889 ± 0.027 | 0.788 ± 0.070 | 0.851 ± 0.041 | **0.898 ± 0.041** | 0.889 ± 0.015 | 0.863 ± 0.047 |
| Name-based optimization | mestranol_similarity | 0.618 ± 0.048 | 0.764 ± 0.035 | 0.579 ± 0.022 | 0.627 ± 0.089 | 0.596 ± 0.018 | 0.717 ± 0.104 | **0.972 ± 0.009** |
| | albuterol_similarity | 0.896 ± 0.008 | 0.918 ± 0.026 | 0.874 ± 0.020 | 0.902 ± 0.019 | 0.929 ± 0.005 | 0.968 ± 0.003 | **0.985 ± 0.024** |
| | thiothixene_rediscovery | 0.534 ± 0.013 | 0.562 ± 0.028 | 0.479 ± 0.025 | 0.559 ± 0.027 | 0.508 ± 0.035 | 0.696 ± 0.081 | **0.727 ± 0.052** |
| | celecoxib_rediscovery | 0.716 ± 0.084 | 0.784 ± 0.011 | 0.582 ± 0.057 | 0.728 ± 0.048 | 0.594 ± 0.105 | 0.508 ± 0.017 | **0.864 ± 0.034** |
| | troglitazone_rediscovery | 0.452 ± 0.048 | 0.556 ± 0.052 | 0.377 ± 0.010 | 0.405 ± 0.007 | 0.381 ± 0.025 | 0.390 ± 0.044 | **0.562 ± 0.019** |
| | perindopril_mpo | 0.537 ± 0.016 | 0.598 ± 0.008 | 0.538 ± 0.009 | 0.493 ± 0.011 | 0.554 ± 0.037 | **0.738 ± 0.016** | 0.600 ± 0.031 |
| | ranolazine_mpo | 0.760 ± 0.009 | **0.802 ± 0.003** | 0.728 ± 0.040 | 0.735 ± 0.013 | 0.725 ± 0.040 | 0.749 ± 0.012 | 0.769 ± 0.022 |
| | sitagliptin_mpo | 0.021 ± 0.003 | 0.479 ± 0.039 | 0.433 ± 0.075 | 0.186 ± 0.055 | 0.548 ± 0.065 | 0.506 ± 0.100 | **0.584 ± 0.067** |
| | amlodipine_mpo | 0.642 ± 0.044 | 0.686 ± 0.046 | 0.625 ± 0.040 | 0.552 ± 0.025 | 0.674 ± 0.018 | **0.776 ± 0.038** | 0.773 ± 0.037 |
| | fexofenadine_mpo | 0.769 ± 0.009 | 0.686 ± 0.010 | 0.779 ± 0.025 | 0.745 ± 0.009 | 0.789 ± 0.016 | 0.773 ± 0.017 | **0.847 ± 0.018** |
| | osimertinib_mpo | 0.834 ± 0.046 | **0.856 ± 0.013** | 0.808 ± 0.012 | 0.762 ± 0.029 | 0.823 ± 0.007 | 0.817 ± 0.016 | 0.835 ± 0.024 |
| | zaleplon_mpo | 0.347 ± 0.049 | 0.438 ± 0.082 | 0.456 ± 0.007 | 0.272 ± 0.026 | 0.475 ± 0.018 | 0.465 ± 0.026 | **0.510 ± 0.031** |
| | median1 | **0.372 ± 0.015** | 0.335 ± 0.012 | 0.287 ± 0.008 | 0.325 ± 0.012 | 0.298 ± 0.019 | 0.338 ± 0.033 | 0.352 ± 0.024 |
| | median2 | 0.294 ± 0.006 | 0.290 ± 0.006 | 0.229 ± 0.017 | **0.308 ± 0.034** | 0.251 ± 0.031 | 0.259 ± 0.019 | 0.275 ± 0.045 |
| Structure-based optimization | isomers_c7h8n2o2 | 0.842 ± 0.029 | 0.954 ± 0.033 | 0.949 ± 0.036 | 0.662 ± 0.071 | 0.948 ± 0.036 | 0.928 ± 0.038 | **0.984 ± 0.008** |
| | isomers_c9h10n2o2pf2cl | 0.642 ± 0.054 | 0.830 ± 0.016 | 0.719 ± 0.047 | 0.469 ± 0.180 | 0.871 ± 0.039 | 0.873 ± 0.019 | **0.874 ± 0.053** |
| | deco_hop | 0.666 ± 0.044 | 0.688 ± 0.060 | 0.619 ± 0.004 | 0.629 ± 0.018 | 0.613 ± 0.016 | 0.827 ± 0.093 | **0.942 ± 0.013** |
| | scaffold_hop | 0.560 ± 0.019 | 0.565 ± 0.008 | 0.517 ± 0.007 | 0.548 ± 0.019 | 0.527 ± 0.019 | 0.559 ± 0.102 | **0.971 ± 0.004** |
| | valsartan_smarts | 0.000 ± 0.000 | 0.000 ± 0.000 | 0.000 ± 0.000 | 0.000 ± 0.000 | 0.000 ± 0.000 | 0.000 ± 0.000 | **0.867 ± 0.092** |
| | Total (↑) | 14.036 | 15.356 | 13.823 | 13.182 | 14.557 | 15.424 | **17.862** |
| | Rank (↓) | 5 | 3 | 6 | 7 | 4 | 2 | 1 |

## 4.2 EMPIRICAL STUDY

First, we motivate the idea of why incorporating chemistry-aware LLMs in GA pipelines is effective. In Figure 2, we show the fitness distribution of an initial pool of random molecules inhibiting JNK3. We then perform a single round of edits to all molecules in the pool using each LLM and plot the resulting fitness distribution of the edited molecules. We find that the distribution for each LLM shifts to slightly higher fitness values, indicating that LLMs do provide useful modifications. However, the overall objective scores are still low, so single-step editing is not sufficient. We then show the fitness distributions of the populations as the genetic optimization progresses and find that the fitness increases to higher values on average, given the same number of oracle calls. We show the performance of direct LLM querying versus the optimization procedure for additional tasks in Appendix C.1.

The results of single-objective optimization across 23 tasks in PMO are shown in Table 1, reporting the AUC top-10 for each task and the overall rank of each model. We show the performance of additional baselines in Appendix D.1. The results indicate that employing any of the three LLMs we tested as genetic operators improves performance over the default Graph-GA. Notably, MOLLEO (GPT-4) outperforms all models in 15 out of 23 tasks and ranks first overall, demonstrating its utility in molecular generation tasks. MOLLEO (BIOT5) achieves the second-best results out of all the

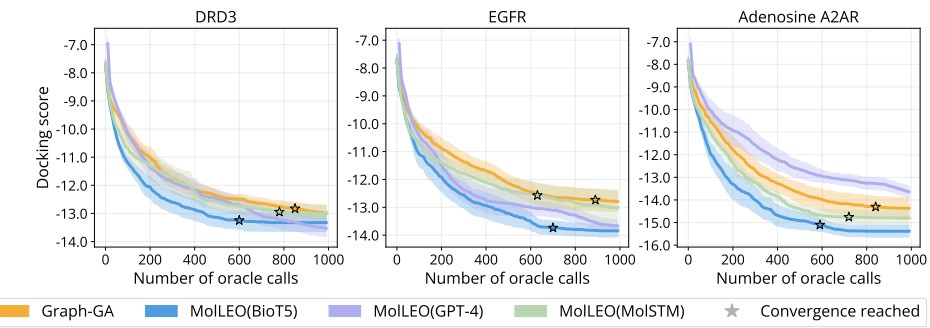

Figure 3: Average docking score of top-10 molecules when docked against DRD3, EGFR, or Adenosine A2A receptor proteins. Lower docking scores are better. For each model, we show the convergence point (the moment of stabilization of the population scores) with a star, if the model converges before 1000 oracle calls have been made. Here, the model is considered to have converged if the mean score of the top 100 molecules does not increase by at least 1e-3 within 5 epochs.

models tested, obtaining a total score close to that of MOLLEO (GPT-4), and has the benefit of being free to use. We observe that MOLLEO (BIOT5) generally performs better than MOLLEO (MOLSTM), producing a higher percentage of molecules with improved fitness after editing, as shown in Appendix C.1. For the tasks `deco_hop` and `scaffold_hop`, there is only a small gain for the open-source MOLLEO models. We speculate that this is because these models have not been trained on molecular descriptions containing SMARTS patterns. Also, it is unclear how well these models perform with negative matching (e.g., `This molecule` **`does not`** `contain the scaffold [#7]-c1n[c;h1]nc2 [c;h1]c(-[#8])[c;h0][c;h1]c12`). We were also interested in knowing whether the open-source models were generating molecules that could have been seen during training. We took ZINC20 (Irwin et al., 2020), a database of 1.4 billion compounds that were used to generate the training set for BioT5, and PubChem (Kim et al., 2023)(∼250K molecules), which was used to generate the training set for MoleculeSTM, and checked if the final molecules for the JNK3 task from each model appeared in the respective datasets. We found that this was not the case; there was no overlap between the generated molecules and the datasets.

We demonstrate empirically that MOLLEO algorithms consistently converge faster than all the considered baselines, i.e., for any given budget of oracle calls, MOLLEO achieves better objective values (see Appendix C.3). This is important when considering how these models can translate to real-world experiments to reduce the number of experiments needed to find ideal candidates. We also study the computational cost of MOLLEO in Appendix D.4.

In Figure 3, we present results for more challenging protein-ligand docking tasks, which better approximate real-world molecular generation scenarios compared to those in Table 1. We plot the average docking scores of the top-10 best molecules for MOLLEO and Graph-GA against the number of oracle calls. We observe that nearly all LLMs in MOLLEO generate molecules with lower (better) docking scores than the baseline model for all three proteins, and they converge faster to the optimal set. Among the three LLMs, MOLLEO (BIOT5) achieves the best performance. Surprisingly, MOLLEO (GPT-4) performs worse than Graph-GA in the Adenosine A2A receptor docking task. In practice, better docking scores and faster convergence rates could result in requiring fewer bioassays to screen molecules, making the process both more cost- and time-effective. We visualize the top-10 molecules found by MOLLEO in `EGFR docking` and `deco_hop` tasks in Appendix D.8.

In Table 2, we show the results of our multi-objective optimization for three tasks. Tasks 1 and 2 are inspired by goals in drug discovery and aim for simultaneous optimization of three objectives: maximizing a molecule's QED, minimizing its synthetic accessibility (SA) score (meaning that it is easier to synthesize), and maximizing its binding score to either JNK3 (Task 1) or GSK3$\beta$ (Task 2). Task 3 is more challenging as it targets five objectives simultaneously: maximizing QED and JNK3 binding, as well as minimizing GSK3$\beta$ binding, DRD2 binding, and SAScore. We find that MOLLEO (GPT-4) consistently outperforms the baseline Graph-GA in all three tasks in terms of hypervolume and summation. In Table 2, we see that the performance of open-source LLMs degrades when introducing multiple objectives into the prompt. We speculate that this performance drop may come from their inability to capture large, information-dense contexts. We further assess both the

Table 2: Summation and hypervolume scores of multi-objective tasks. We report the results for two aggregation methods: Summation (Sum) and Pareto optimality (PO). Sum(AUC) refers to the summation of top-10 AUC for all optimized objectives. The best results for each task are bolded.

| Aggregate objective | Model | Task 1: QED (↑), JNK3 (↑), SAscore (↓) | | Task 2: QED (↑), GSK3β (↑), SAscore (↓) | | Task 3: QED (↑), JNK3 (↑), SAscore (↓),GSK3β (↓), DRD2 (↓) | |
|---|---|---|---|---|---|---|---|
| | | Sum(AUC) | Hypervolume | Sum(AUC) | Hypervolume | Sum(AUC) | Hypervolume |
| Sum | Graph-GA | 1.967 ± 0.088 | 0.713 ± 0.083 | 2.186 ± 0.069 | 0.719 ± 0.055 | 3.856 ± 0.075 | 0.162 ± 0.048 |
| | MOLLEO (MOLSTM) | 2.177 ± 0.178 | 0.625 ± 0.162 | 2.349 ± 0.132 | 0.303 ± 0.024 | **4.040 ± 0.097** | 0.474 ± 0.193 |
| | MOLLEO (BIOT5) | 1.946 ± 0.222 | 0.592 ± 0.199 | 2.306 ± 0.120 | 0.693 ± 0.093 | 3.904 ± 0.092 | 0.266 ± 0.201 |
| | MOLLEO (GPT-4) | **2.367 ± 0.044** | **0.752 ± 0.085** | **2.543 ± 0.014** | **0.832 ± 0.024** | 4.017 ± 0.048 | **0.606 ± 0.086** |
| PO | Graph-GA | 2.120 ± 0.159 | 0.603 ± 0.082 | 2.339 ± 0.139 | 0.640 ± 0.034 | 4.051 ± 0.155 | 0.606 ± 0.052 |
| | MOLLEO (MOLSTM) | 2.234 ± 0.246 | 0.472 ± 0.248 | 2.340 ± 0.254 | 0.202 ± 0.054 | 3.989 ± 0.145 | 0.381 ± 0.204 |
| | MOLLEO (BIOT5) | 2.325 ± 0.164 | 0.630 ± 0.120 | 2.299 ± 0.203 | 0.645 ± 0.127 | 3.946 ± 0.115 | 0.367 ± 0.177 |
| | MOLLEO (GPT-4) | **2.482 ± 0.057** | **0.727 ± 0.038** | **2.631 ± 0.023** | **0.820 ± 0.024** | **4.212 ± 0.034** | **0.696 ± 0.029** |

| Model | JNK3 Top-10 AUC |
|---|---|
| Initial fitness | 0.373±0.079 |
| Graph-GA | 0.787±0.035 |
| MOLLEO (MOLSTM) | 0.815±0.048 |
| MOLLEO (BIOT5) | 0.799±0.036 |
| MOLLEO (GPT-4) | **0.844±0.052** |

Table 3: Initializing MOLLEO with the best molecules from ZINC 250K (Sterling & Irwin, 2015). The results of three different LLMs in MOLLEO and Graph-GA are compared. For all molecules in ZINC 250K, we run the JNK3 oracle and select the top 120 molecule pool. We run MOLLEO initializing from this pool of molecules and optimizing JNK3. We report the top-10 AUC on the output of MOLLEO.

structural and objective diversity of the Pareto optimal sets across all tasks, and we visualize these sets in objective space for MOLLEO and Graph-GA on Tasks 1 and 2 (see Appendix D.7).

Given that the goal of EAs is to improve upon the properties of an initial pool of molecules and discover new molecules, we showcase these abilities by generating a set of molecules with higher objective values than the best-known molecules from ZINC 250K (Sterling & Irwin, 2015). That is, we initialize the molecular pool with the best molecules from ZINC 250K and run the optimization with MOLLEO and Graph-GA. We report the top-10 AUC on the JNK3 task in Table 3 and find that MOLLEO algorithms are consistently able to outperform the baseline model and improve upon the best values found in the existing dataset. We briefly investigate the use of retrieval augmented search in Appendix C.5 and find that incorporating information from existing databases is helpful. To further validate the effectiveness of the LLM-based genetic operators, we compare the molecules before and after LLMs' editing in Appendix D.5 and check whether the optimization objectives are in the open-source LLMs training data in Appendix D.6. We also incorporate MOLLEO into other GAs and generative models to validate its generalization capability in Appendix D.2 and Appendix D.3.

## 5 CONCLUSION

Herein, we propose MOLLEO: the first demonstration of incorporating LLMs into evolutionary algorithms for molecular discovery. We show that chemistry-aware LLMs can serve as informed proposal generators, resulting in superior optimization performance across multiple molecular optimization benchmarks, including protein-ligand docking. Furthermore, we show that both open-source and commercial versions of MOLLEO can be used in scenarios that involve numerous objective evaluations and can generate higher-ranked candidates with fewer evaluation calls compared to baseline models. Because the structural perturbations of MOLLEO are more effective than random perturbations in a genetic algorithm, it will become more feasible to deploy oracles that are computationally more expensive but more accurate in representing the target property, generating candidates that show greater promise for real-life applications. This is an important consideration due to the high experimental costs of testing candidates. As LLMs continue to advance, we anticipate that the performance of the MOLLEO framework will also continue to improve, making MOLLEO a promising tool for applications in generative chemistry. We introduce the future work in Appendix A.2.

## 6 REPRODUCIBILITY STATEMENT

Our code is available at `https://github.com/zoom-wang112358/MOLLEO`. We provide the experimental details and the choice of hyperparameters in Section 4.1 and Appendix B. The pseudocode of MOLLEO algorithm is in Algorithm 1.

## 7 ACKNOWLEDGEMENTS

M.S. thanks Ella Rajaonson for feedback on protein-ligand docking and Austin Cheng for discussions on molecule generation. Y.D. thanks Delia Qu for helpful discussions on evolutionary algorithms. A. A.-G. thanks Dr. Anders G. Frøseth for his generous support. A. A.-G. also acknowledges the generous support of Natural Resources Canada and the Canada 150 Research Chairs program.

This work was supported in part by NSF IIS-2008334, IIS-2106961, CAREER IIS-2144338, ONR MURI N00014-17-1-2656, and computing resources from Microsoft Azure. Resources used in preparing this research were also provided, in part, by the Vector Institute and the Acceleration Consortium.

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

# Appendix

## A EXTENDED DESCRIPTIONS

### A.1 EXTENDED RELATED WORK

**Benchmarking LLMs on Chemistry Tasks**  ChemLLMBench benchmarked several widely-used LLMs on a set of eight chemistry tasks, such as property prediction, reaction prediction, and molecule captioning (Guo et al., 2023b). The results showed that while LLMs can perform well in selection tasks, they struggle with tasks requiring more in-depth chemical reasoning, such as property-conditioned generation. This motivates the need to improve how LLMs are used in generative tasks. Similarly, SciBench evaluated LLMs on free-response college-level exam questions across various science disciplines, including chemistry, which required complex, multi-step solutions (Wang et al., 2023). Their results indicated that LLMs were unable to generate correct solutions for the majority of questions (Wang et al., 2023). However, progress of LLMs has been noted in general question-answering capabilities: a recent work introduced ChemBench, a dataset of over 7,000 question-answer pairs aimed at providing a systematic understanding of LLM capabilities across different subdomains in chemistry (Mirza et al., 2024). It was concluded that state-of-the-art LLMs such as GPT-4 and Claude 3 were able to beat human chemists on these questions on average, although they still struggle with physical and commonsense chemical reasoning.

**LLMs and Evolutionary Algorithms**  Previous research has demonstrated that language models can be incorporated as operators in evolutionary algorithms in applications such as code and prompt generation (Lehman et al., 2023). For example, OPRO and LMEA use LLMs to optimize solutions for different mathematical optimization problems (Yang et al., 2024; Liu et al., 2023a). Other works have shown that LLMs can be used as crossover and mutation operators to directly optimize prompts using a training set, outperforming human-engineered prompts (Fernando et al., 2023; Guo et al., 2023a). Other applications of LLMs in evolutionary frameworks have been code synthesis (FunSearch (Romera-Paredes et al., 2024)), generation of reward functions in RL for robot control (Eureka (Ma et al., 2024), and resource allocation in public health settings (Behari et al., 2024).

**Multi-objective optimization frameworks**  In our work, we study the effectiveness of LLM-based mutations in a multi-objective molecular optimization setting. To ensure simplicity and clarity in our evaluation, we adopt straightforward approaches, such as the sum of objectives and Pareto set selection as selection criteria. Classic methods like MOEA/D (Zhang & Li, 2007) and NSGA-III (Deb & Jain, 2013) are designed to handle scenarios where the Pareto set exceeds the population capacity. MOEA/D uses decomposition to assign each solution to a specific subproblem defined by a weight vector. If the size of the Pareto set exceeds the population size, MOEA/D will select solutions based on their contribution to specific subproblems, so that it can ensure a balance between diversity and convergence. NSGA-III uses reference points in the objective space to maintain diversity. When the Pareto front size exceeds the population, a clustering mechanism based on the reference points is applied to select solutions that best represent different regions of the Pareto front. Additionally, there are also some recent works focusing on this topic. For example, Sun et al. (2022) developed a Monte Carlo tree search algorithm that evaluates rewards by comparing new molecular structures against a maintained global Pareto set. Shin et al. introduced a method to decompose the optimization process into a progressive sequence based on the order of objectives. Zhu et al. (2024) integrated GFlowNets with a preference-conditioned sum of objective functions, further advancing the optimization landscape.

### A.2 FUTURE WORK

Molecular discovery and design is a rich field with numerous practical applications, many of which extend beyond the current study's scope but remain relevant to the proposed framework. Integrating LLMs into evolutionary algorithms offers versatility through plain text specifications, suggesting that the MOLLEO framework can be applied to scenarios such as drug discovery, expensive *in silico* simulations, and the design of materials or large biomolecules. Future work will aim to further improve the quality of proposed candidates, both in terms of their objective values and the speed with which they are found.

### A.3 COMPUTATIONAL RESOURCES

Our experiments were computed on NVIDIA A100-SXM4-80GB and T4V2 GPUs. Some of our experiments utilized the GPT-4 model; this refers to the `gpt-4-turbo` checkpoint from 2023-07-01 [2]. All GPT-4 checkpoints were hosted on Microsoft Azure[3].

### A.4 LIMITATIONS

All benchmarks and tasks evaluated in this study are proxies for real chemical properties and may not correctly capture the true chemical performance of molecules in the real world. Thus, the effectiveness of our model in real-world applications remains to be thoroughly validated.

### A.5 BROADER IMPACT

The methods proposed in this paper aim to find compounds with desired properties more efficiently, which can benefit many areas, including drug discovery and materials design. While we do not foresee negative societal impacts from our methods, we acknowledge the potential of their dual use for nefarious purposes. We encourage discussions around these issues and strongly support the development and deployment of safeguards to prevent them.

## B HYPERPARAMETERS AND ADDITIONAL EXPERIMENTAL DETAILS

For the choice of hyperparameters, we use the best practices from Graph-GA (Jensen, 2019), the baseline genetic algorithm that we build our method upon. We kept the best hyperparameters that were determined in (Gao et al., 2022). In each iteration, Graph-GA samples two molecules with a probability proportional to their fitnesses for crossover and mutation and then randomly mutates the offspring with probability $p_m = 0.067$. This process is repeated to generate 70 offspring. The fitnesses of the offspring are measured, and the top 120 most fit molecules in the entire pool are kept for the next generation. For docking experiments, we reduce the number of generated offspring to 7 and the population size to 12 due to long experiment runtimes. We set the maximum number of oracle calls to 10,000 for all experiments except docking, where we set it to 1,000. We kept the default early-stopping criterion the same as in PMO (Gao et al., 2022), which is that we terminate the algorithm if the mean score of the top 100 molecules does not increase by at least 1e-3 within five epochs.

In the multi-objective optimization tasks, we applied a simple transformation by using $1 - score$ for the objectives involving minimization. Also, we ensure all objectives remain within the range of 0 to 1 by using normalization. This approach allows for consistent scalarization and comparability across objectives.

MOLLEO (MOLSTM) involves additional hyperparameters when doing gradient descent; we investigate their selection in Appendix C.4. Additionally, we investigate design choices for MOLLEO (GPT-4) in Appendix C.5. We use three tasks for model development: JNK3, `perindopril_mpo`, and `isomers_c9h10n2o2pf2cl`; the rest are only evaluated during test-time. For each model, we show the prompts we used in Appendix E. We created prompts similar to those demonstrated in the original source code of each model, replacing each template with a task description. We briefly investigate the impact of prompt selection in Appendix C.6.

All experiments are conducted with five random seeds. The computational resources we utilized are described in Appendix A.3.

---

[2].https://platform.openai.com/docs/models
[3] *.openai.azure.com

Table A1: Viability of LLM edits. We prompt different LLMs with descriptions of JNK3 and perindopril_mpo target objectives on an initial random pool of molecules drawn from 5 random seeds. We report the percentage of valid molecules (number of valid molecules/number of total molecules), the percentage of molecules with higher fitness after editing, and the mean fitness increase of those molecules.

| **Metric** | MoleculeSTM | BioT5 | GPT-4 |
|---|---|---|---|
| Percent valid molecules | peridopril_mpo: 0.938 JNK3: 0.928 | peridopril_mpo: 1.000 JNK3: 1.000 | peridopril_mpo: 0.862 JNK3: 0.835 |
| Percent molecules with higher fitness after editting | peridopril_mpo: 0.456 JNK3: 0.206 | peridopril_mpo: 0.568 JNK3: 0.513 | peridopril_mpo: 0.240 JNK3: 0.263 |
| Mean fitness increase | peridopril_mpo: +0.033 JNK3: +0.022 | peridopril_mpo: +0.208 JNK3: +0.0320 | peridopril_mpo: +0.032 JNK3: +0.0262 |

## C  ABLATION STUDIES

### C.1  PERFORMANCE OF SINGLE-STEP MOLECULE EDITING

To motivate the incorporation of LLMs into a GA framework, we directly query the LLMs we consider to edit a molecule towards a certain property and calculate: (1) the percentage of valid molecules that are output (given that not all SMILES are valid molecules) and (2) which of the output molecules have higher fitness. We show these results on the JNK3 inhibition task in Table A1 and find that MolSTM and GPT-4 are not always able to produce valid molecules, whereas BioT5 always is due to its use of SELFIES. We also found that BioT5 produced more molecules with higher fitness values compared to the other LLMs.

In Table A3, we show the performance of directly querying LLMs with an initial pool of molecules on additional tasks. We find that while LLMs are able to edit the molecule pool to improve fitness marginally, using them in an optimization framework results in much better fitness values.

### C.2  INCORPORATING LLM-BASED GENETIC OPERATORS INTO GRAPH-GA

There are many ways to incorporate LLMs as genetic operators in a GA framework. We investigate several options. First, we investigate using LLMs as a crossover operator. For GPT-4 and BioT5, we gave each model two parent molecules as input and a description of the objective, and asked the model to produce a molecule as an output. Because MolSTM aligns molecule embeddings with text embeddings, our crossover operation was to either take a linear or spherical interpolation of the parent molecule embeddings and maximize the similarity of the resulting embedding to the text objective. For the mutation operator, we prompted each LLM with a molecule and a description of the objective. Finally, we investigated the impact of applying a selection pressure in the form of a filter, where we only mutated the top $Y$ molecules and pruned the resulting offspring by distance to the best molecule overall. We show the results for all operator settings we tried in Table A2 and show which operators we ended up using for each LLM in the final framework.

### C.3  OPTIMIZATION TRENDS OVER SINGLE-OBJECTIVE TASKS.

In Figure A1, we show the optimization curves for three tasks: JNK3, perindopril_mpo, and iso­mers_c9h10n2o2pf2cl.

### C.4  MOLECULESTM HYPERPARAMETER SELECTION

MolSTM has several hyperparameters; in this section, we motivate our choices for the final model. The first is the number of population members that are selected to undergo LLM-based mutations

Table A2: **Top-10 AUC on 5 random seeds for the JNK3 and perindopril_mpo tasks using different combinations of genetic operators.** The operators used for each model to compute the final results in the main paper are indicated with a ✅ symbol.

| Operators | Graph-GA (Baseline) | MOLLEO (MOLSTM) | MOLLEO (BIOT5) | MOLLEO (GPT-4) |
|---|---|---|---|---|
| (Default Graph-GA settings) CROSSOVER: Random MUTATION: Random, $p_m = 0.067$ | peridopril_mpo: 0.538 ± 0.009 JNK3: 0.553 ± 0.136 ✅ | N/A | N/A | N/A |
| CROSSOVER: LLM MUTATION: Random, $p_m = 0.067$ | N/A | peridopril_mpo: 0.499 ± 0.012 [linear] 0.505 ± 0.018 [spherical] JNK3: 0.722±0.046 [linear] 0.744 ± 0.055 [spherical] | peridopril_mpo: 0.727 ± 0.013 JNK3: 0.436 ± 0.052 | peridopril_mpo: 0.600 ± 0.031 JNK3: 0.790 ± 0.027 ✅ |
| CROSSOVER: Random MUTATION: LLM, $p_m = 0.067$ | N/A | peridopril_mpo: 0.532 ± 0.034 JNK3: 0.631 ± 0.327 | peridopril_mpo: 0.676 ± 0.034 JNK3: 0.650 ± 0.096 | peridopril_mpo: 0.552 ± 0.024 JNK3: 0.673 ± 0.047 |
| CROSSOVER: Random MUTATION: LLM, $p_m = 1$ | N/A | peridopril_mpo: 0.513 ± 0.040 JNK3: 0.553 ± 0.193 | peridopril_mpo: 0.686 ± 0.343 JNK3: 0.708 ± 0.030 | peridopril_mpo: 0.615 ± 0.058 JNK3: 0.762 ± 0.044 |
| CROSSOVER: Random MUTATION: Selected top $Y$ molecules, randomly mutated, pruned offspring by distance to top-1 molecule | peridopril_mpo: 0.579 ± 0.044 JNK3: 0.571 ± 0.109 | N/A | N/A | N/A |
| CROSSOVER: Random MUTATION: Selected top $Y$ molecules, mutated with LLM, pruned offspring by distance to top-1 molecule | N/A | peridopril_mpo: 0.554 ± 0.034 JNK3: 0.730 ± 0.188 ✅ | peridopril_mpo: 0.740 ± 0.032 JNK3: 0.728 ± 0.079 ✅ | peridopril_mpo: 0.575 ± 0.074 JNK3: 0.758 ± 0.031 |
| CROSSOVER: LLM MUTATION: Selected top $Y$ molecules, mutated with LLM, pruned offspring by distance to top-1 molecule | N/A | peridopril_mpo: 0.490 ± 0.016 [linear] 0.517 ± 0.006 [spherical] JNK3: 0.692 ± 0.110 [linear] | peridopril_mpo: 0.736 ± 0.014 JNK3: 0.429 ± 0.110 | peridopril_mpo: 0.592 ± 0.035 JNK3: 0.794 ± 0.026 |

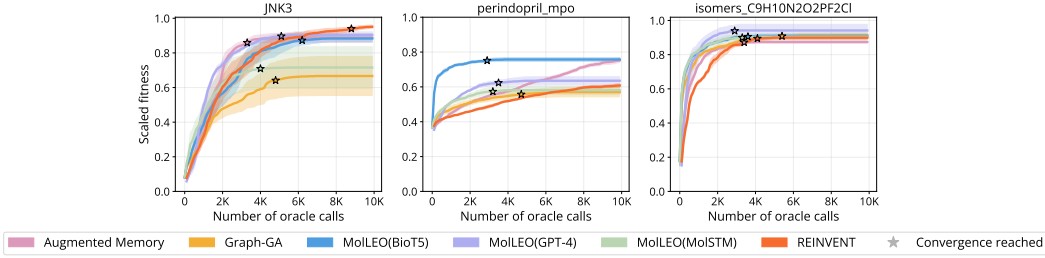

Figure A1: Average of top-10 molecules generated by MoLLEO and Graph-GA models for three tasks over an increasing number of oracle calls. For each model, we show the convergence point with a star. The model is considered to have converged if the mean score of the top 100 molecules does not increase by at least 1e-3 within five epochs.

Table A3: Ablation studies of LLM editing based on direct user queries. Top-10 average objective scores are reported.

|  | JNK3 | isomers_c9h10n2o2pf2cl | perindopril_mpo |
|---|---|---|---|
| Initial population | 0.085 ± 0.010 | 0.101 ± 0.025 | 0.281 ± 0.026 |
| MolSTM - direct query | 0.084 ± 0.008 | 0.201 ± 0.040 | 0.390 ± 0.008 |
| MoLLEO (MoLSTM) | **0.716 ± 0.240** | **0.905 ± 0.0372** | **0.572 ± 0.041** |
| BioT5 - direct query | 0.109 ± 0.012 | 0.260 ± 0.076 | 0.648 ± 0.019 |
| MoLLEO (BioT5) | **0.883 ± 0.040** | **0.909 ± 0.015** | **0.759 ± 0.019** |
| GPT-4 - direct query | 0.164 ± 0.076 | 0.686 ± 0.127 | 0.388 ± 0.075 |
| MoLLEO (GPT-4) | **0.926 ± 0.052** | **0.935 ± 0.048** | **0.643 ± 0.094** |

(Algorithm 1). In Table A4, we show the Top-10 AUC after choosing different numbers of top-scoring candidates for editing by MoleculeSTM. We found that 30 candidates resulted in the best performance. Note that we used a different prompt for this experiment than the one used to obtain results in Table 1 (see Appendix C.6). We use 30 candidates anytime the filter is employed for all models, although this hyperparameter can be ablated independently for each model.

MoleculeSTM has several hyperparameters related to molecule generation since it involves gradient descent to optimize an input molecule embedding based on a text prompt. We look at two hyperparameters, the number of gradient descent steps (epochs) and learning rate, and plot the results in Figure A2. We find that if the learning rate is too large (lr=1), the mean fitness changes unpredictably, but if it is too small (lr=1e-2), there are minimal changes to the mean fitness. Setting the learning rate to 1e-1 results in more consistent improvements in mean fitness. We also set the number of epochs to 30 since more epochs are too time-consuming and fewer do not result in noticeable fitness changes.

## C.5 GPT-4 ABLATIONS

We conduct experiments to understand the performance of MoLLEO (GPT-4) in the following settings: different numbers of offspring in each generation, different underlying GPT models, incorporating retrieval augmentation methods, and different rules from Graph-GA and SMILES-GA in Table A5 and Table A6, and describe the results in following sections.

| Number of top-scoring candidates selected for mutation | Top-10 AUC |
|---|---|
| 20 | 0.680±0.213 |
| 30 | **0.730±0.188** |
| 50 | 0.627±0.250 |

Table A4: Top-10 AUC on JNK3 binding task with varying numbers of top-scoring candidates selected to undergo LLM-based mutations.

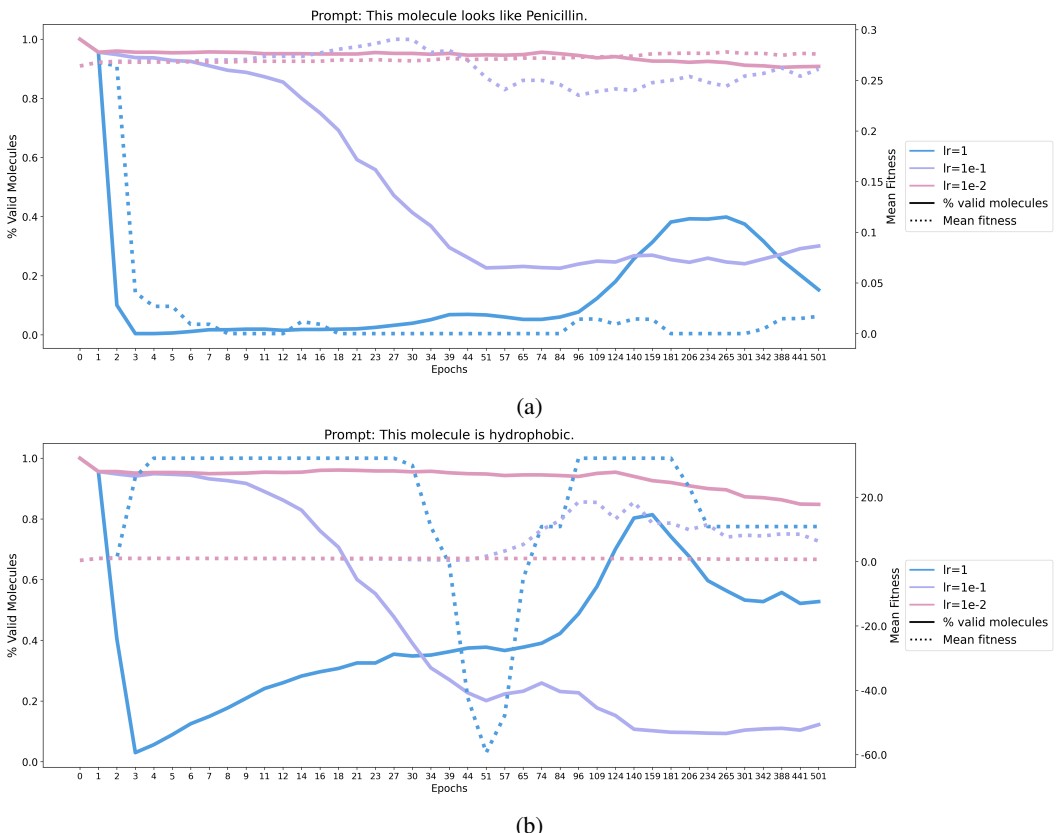

Figure A2: Mean fitness and percent valid molecules with a varying number of gradient descent epochs (plotted on log-scale) and learning rates in MoleculeSTM on two tasks: (a) molecular similarity to Penicillin (based on Tanimoto distance) and (b) molecule hydrophobicity (logP).

Table A5: Ablation study on MOLLEO (GPT-4). Impact of the number of offspring in each round and retrieval-augmented search (RAG).

| | Number of offspring | | | RAG Search | |
|---|---|---|---|---|---|
| | 20 | 70 | 200 | w. RAG | w/o. RAG |
| jnk3 | 0.731±0.012 | 0.790±0.027 | 0.785±0.022 | 0.830±0.047 | 0.790±0.027 |
| isomer_c9h10n2o2pf2cl | 0.967±0.010 | 0.874±0.053 | 0.960±0.049 | 0.982±0.018 | 0.874±0.053 |
| perindopril mpo | 0.573±0.042 | 0.600±0.031 | 0.580±0.028 | 0.717±0.024 | 0.600±0.031 |

Table A6: Ablation study on MOLLEO (GPT-4). Impact of different versions of LLMs and rules from different sources.

| | Different Versions of LLMs | | Rules | | |
|---|---|---|---|---|---|
| | GPT-3.5 | GPT-4 | No rules | Graph-GA rules | SMILES-GA rules |
| jnk3 | 0.669±0.104 | 0.790±0.027 | 0.765±0.047 | 0.790±0.027 | 0.774±0.084 |
| isomer_c9h10n2o2pf2cl | 0.902±0.021 | 0.874±0.053 | 0.871±0.085 | 0.874±0.053 | 0.872±0.029 |
| perindopril mpo | 0.564±0.022 | 0.600±0.031 | 0.562±0.042 | 0.600±0.031 | 0.583±0.031 |

**Number of offspring**   We vary the number of offspring generated in each iteration of MOLLEO (GPT-4) on three tasks and find that 70 offspring produces, on average, the best results, which is also the same number determined in (Gao et al., 2022)

**Retrieval-augmented search**   To explore how retrieval can enhance LLMs in the optimization process, we incorporate a retrieval-augmented search module into MOLLEO (GPT-4). Specifically, after offspring are proposed, 1,000 molecules are randomly sampled from ZINC 250K. From these, 20 molecules are selected based on their Tanimoto similarity to the top 20 molecules in the current population. These retrieved molecules then replace the 20 worst molecules in the population. In Table A5, the results show that this approach is effective in improving the optimization results of MOLLEO (GPT-4) for each task.

**GPT-3.5 vs. GPT-4**   We tested MOLLEO using both GPT-4 and GPT-3.5, an older version of the model. In Table A6, we show that GPT-4 outperforms GPT-3.5 on two tasks, although GPT-3.5 still beats the baseline Graph-GA algorithm (Table 1). Interestingly, GPT-3.5 beats GPT-4 on a task based on structure-based optimization.

**Different rules**   In Graph-GA, the default crossover and mutation operators are pre-defined by domain experts based on chemical knowledge. These pre-defined operators can be considered as rules guiding the generation process. Here also consider rules from another source, SMILES-GA (Yoshikawa et al., 2018), which defines rules that operate on SMILES strings instead of graphs. To evaluate the impact of rules from different sources, we perform an ablation study on MOLLEO and also conduct experiments without any rules, where LLMs are repeatedly queried to propose molecules until the offspring size reaches the target number in each round. The results shown in Table A6 indicate that both Graph-GA and SMILES-GA rules are better than not using results at all, and Graph-based rules are better than SMILES-based rules.

### C.5.1   GPT-4 IN AN ACTIVE LEARNING FRAMEWORK

We investigate the performance of GPT-4 when the EA framework is replaced with an active learning setting. This can be thought of as testing the impact of the genetic operators in the underlying genetic framework. In this setting, we initialize a population pool and randomly sample $k$ molecules from the pool. We then pass the molecules to GPT-4 and query it for a new molecule with better objective values. After generating a batch of molecules, we integrate the batch back into the population without selection, allowing the population to grow until it reaches the budget of oracle calls. In our experiment, we set the budget to 10,000 oracle calls, the batch size to 100, and $k$ to 2.

The results, shown in Table A7, indicate that the active learning setting achieves subpar performance compared to MOLLEO (GPT-4). This demonstrates that while LLMs like GPT-4 can modify existing molecules, they struggle to independently propose high-quality molecules, underscoring the necessity of the evolutionary process. Interestingly, we observe that the active learning setting performs relatively well on the `isomer` task compared to the other two; this can maybe be attributed to the `isomer` task being simple.

Table A7: Ablation studies of active learning (AL) on GPT-4. We report the Top-10 AUC of single objective results.

|  | GPT4-AL | MOLLEO (GPT-4) |
|---|---|---|
| JNK3 | 0.583±0.042 | 0.790±0.027 |
| isomer_c9h10n2o2pf2cl | 0.873±0.048 | 0.874±0.053 |
| perindopril mpo | 0.539±0.046 | 0.600±0.031 |

## C.6 IMPACT OF PROMPT SELECTION

The choice of prompt for a given task is an important consideration, as some prompts can be better aligned with information the model knows. For example, the prompt we used in MOLLEO (MOLSTM) for the JNK3 inhibition task was "`This molecule inhibits JNK3.`" However, there are multiple ways of describing inhibition and multiple ways of identifying the enzyme (JNK3, c-Jun N-terminal kinase 3). To that end, we investigate the impact of prompt selection on downstream performance.

To generate a set of prompts, we prompted GPT-4 to generate ten synonymous phrases for an input prompt. We then computed the Spearman rank-order correlation coefficient (Spearman's $\rho$) of each phrase on an initial molecule pool between the cosine similarity generated by MoleculeSTM and the ground truth fitness values. Finally, we ran the genetic optimization using MOLLEO (MOLSTM) with the input prompt and the prompt with the highest Spearman rank-order correlation coefficient.

On the JNK3 task, the default prompt we wrote was "`This molecule inhibits JNK3.`", which had a Spearman's $\rho$ of -0.0161. The prompt with the largest Spearman's $\rho$ (0.1202) was "`This molecule acts as an antagonist to JNK3.`" When we ran MOLLEO (MOLSTM) with the default input prompt, the top-10 AUC was 0.643 ± 0.226. When we ran MOLLEO (MOLSTM) using the prompt with the largest Spearman's $\rho$, the top-10 AUC was 0.730 ± 0.188. This demonstrates that prompt selection can influence downstream results, especially for smaller models, and opens the door for future work in this area.

Table A8: Top-10 AUC of single-objective tasks on additional baseline models. The best model for each task is bolded and the top three are underlined.

| Task type | Method objective (↑) | DST | REINVENT | Augmented Memory | Graph GA | GP BO | MOLLEO (MolSTM) | MOLLEO (BioT5) | MOLLEO (GPT-4) |
|---|---|---|---|---|---|---|---|---|---|
| Property optimization | QED | 0.939 ± 0.000 | 0.941 ± 0.000 | 0.941 ± 0.000 | 0.940 ± 0.000 | 0.937 ± 0.000 | 0.937 ± 0.002 | 0.937 ± 0.002 | **0.948 ± 0.000** |
|  | JNK3 | 0.677 ± 0.157 | 0.783 ± 0.023 | 0.773 ± 0.073 | 0.553 ± 0.136 | 0.564 ± 0.155 | 0.643 ± 0.226 | 0.728 ± 0.079 | **0.790 ± 0.027** |
|  | GSK3$\beta$ | 0.767 ± 0.103 | 0.865 ± 0.043 | 0.889 ± 0.027 | 0.788 ± 0.070 | 0.851 ± 0.041 | **0.898 ± 0.041** | 0.889 ± 0.015 | 0.863 ± 0.047 |
| Name-based optimization | mestranol_similarity | 0.435 ± 0.015 | 0.618 ± 0.048 | 0.764 ± 0.035 | 0.579 ± 0.022 | 0.627 ± 0.089 | 0.596 ± 0.018 | 0.717 ± 0.104 | **0.972 ± 0.009** |
|  | albuterol_similarity | 0.614 ± 0.021 | 0.896 ± 0.008 | 0.918 ± 0.026 | 0.874 ± 0.020 | 0.902 ± 0.019 | 0.929 ± 0.005 | 0.968 ± 0.003 | **0.985 ± 0.024** |
|  | thiothixene_rediscovery | 0.352 ± 0.011 | 0.534 ± 0.013 | 0.562 ± 0.028 | 0.479 ± 0.025 | 0.559 ± 0.027 | 0.508 ± 0.035 | 0.696 ± 0.081 | **0.727 ± 0.052** |
|  | perindopril_mpo | 0.470 ± 0.015 | 0.537 ± 0.016 | 0.598 ± 0.008 | 0.538 ± 0.009 | 0.493 ± 0.011 | 0.554 ± 0.037 | **0.738 ± 0.016** | 0.600 ± 0.031 |
|  | ranolazine_mpo | 0.665 ± 0.010 | 0.760 ± 0.009 | **0.802 ± 0.003** | 0.728 ± 0.012 | 0.735 ± 0.013 | 0.725 ± 0.040 | 0.749 ± 0.012 | 0.769 ± 0.022 |
|  | osimertinib_mpo | 0.794 ± 0.007 | 0.834 ± 0.046 | **0.856 ± 0.013** | 0.808 ± 0.012 | 0.762 ± 0.029 | 0.823 ± 0.007 | 0.817 ± 0.016 | 0.835 ± 0.024 |
| Structure-based optimization | isomers_c7h8n2o2 | 0.706 ± 0.033 | 0.842 ± 0.029 | 0.954 ± 0.033 | 0.949 ± 0.036 | 0.662 ± 0.071 | 0.948 ± 0.036 | 0.928 ± 0.038 | **0.984 ± 0.008** |
|  | scaffold_hop | 0.501 ± 0.006 | 0.560 ± 0.019 | 0.565 ± 0.008 | 0.517 ± 0.007 | 0.548 ± 0.019 | 0.527 ± 0.019 | 0.559 ± 0.102 | **0.971 ± 0.004** |
|  | valsartan_smarts | 0.000 ± 0.000 | 0.000 ± 0.000 | 0.000 ± 0.000 | 0.000 ± 0.000 | 0.000 ± 0.000 | 0.000 ± 0.000 | 0.000 ± 0.000 | **0.867 ± 0.092** |

Table A9: Performance of Augmented Memory framework with and without LLM-based genetic operators. We report the Top-10 AUC of single objective results.

| Augmented Memory Objective(↑) | w/o LLM | w/ BioT5 | w/ GPT-4 |
|---|---|---|---|
| JNK3 | 0.773 ± 0.073 | 0.781 ± 0.094 | **0.794 ± 0.087** |
| albuterol_similarity | 0.918 ± 0.026 | 0.925 ± 0.076 | **0.941 ± 0.033** |

Table A10: Performance of Genetic GFN with different genetic operators. We report the Top-10 AUC of single objective results.

| Genetic GFN Objective(↑) | default GA | MolLEO(BioT5) | MolLEO(GPT-4) |
|---|---|---|---|
| JNK3 | 0.766 ± 0.077 | 0.775 ± 0.056 | **0.783 ± 0.034** |
| albuterol_similarity | 0.946 ± 0.013 | 0.962 ± 0.017 | **0.971 ± 0.020** |

# D  EXTENDED EXPERIMENT RESULTS

## D.1  ADDITIONAL BASELINE MODELS

We report the performance of an additional baseline, DST (Fu et al., 2021), on randomly selected tasks in Table A8.

## D.2  INCOPORATING LLMs INTO AUGMENTED MEMORY

To further evaluate the effectiveness of LLM-based genetic operators, we integrate them into a framework with augmented memory mechanisms (Guo & Schwaller, 2024) to refine the molecules stored in a replay buffer. The results presented in Table A9 demonstrate that incorporating both BioT5 and GPT-4 into this framework improves performance, indicating the capability of LLM-based genetic operators to effectively augment molecules in the replay buffer. The performance improvement is not large, which is likely due to the application of only a single round of edits.

## D.3  INCOPORATING LLMs INTO GENETIC GFN AND JANUS

We experiment with extending the MOLLEO framework to other genetic algorithms other than Graph GA including Genetic GFN (Kim et al., 2024) and JANUS (Nigam et al., 2021). Genetic GFN combines GAs with GFlowNet by first sampling molecules using the current policy during the GFlowNet training stage. These sampled molecules are then refined into higher-reward ones using GAs, after which the policy is fine-tuned using the refined samples. We replace the default GA in Genetic GFN with MOLLEO and the results are shown in Table A10. Both MOLLEO (BIOT5) and MOLLEO (GPT-4) outperform the default GA, which relies on predefined rules crafted by chemical experts.

JANUS maintains two distinct populations of molecules that can exchange members, each governed by specialized genetic operators—one set focused on exploration and the other on exploitation. The exploitative genetic operators apply molecular similarity as an additional selection pressure, while the explorative operators leverage guidance from a deep neural network (DNN) trained on molecules across all previous generations. We replace these genetic operators with MOLLEO (MOLSTM) and MOLLEO (BIOT5). The results of this approach are presented in Table A11. The results show that MOLLEO can edit the molecules effectively in both Genetic GFN and JANUS, indicating the utility of LLM-based genetic operators in several settings.

## D.4  COMPUTATIONAL COST OF MOLLEO

The most expensive part of chemistry experiments is often the oracle call. In table A12, we show the computational costs of MOLLEO vs Graph GA for two experiments: JNK3, which uses a lightweight oracle, and docking against human dopamine D3 receptor (PDB ID 3pbl), which is more expensive.

Table A11: Performance of JANUS with different genetic operators. We report the Top-10 AUC of single objective results.

| JANUS Objective($\uparrow$) | default GA | MOLLEO (MOLSTM) | MOLLEO (BIOT5) |
|---|---|---|---|
| JNK3 | $0.678 \pm 0.031$ | $0.680 \pm 0.024$ | $\mathbf{0.685 \pm 0.044}$ |
| albuterol_similarity | $0.712 \pm 0.049$ | $0.779 \pm 0.038$ | $\mathbf{0.904 \pm 0.051}$ |

Table A12: Average Running time of five seeds of MOLLEO framework

| Task | Model | Avg time |
|---|---|---|
| Docking 3pbl | Graph GA | 8h 23m 37s $\pm$ 1h 46m |
| | MOLLEO (BIOT5) | 6h 57m 14s $\pm$ 1h 24m |
| | MOLLEO (GPT-4) | 12h 14m 25s $\pm$ 2h 33m |
| JNK3 | Graph GA | 12m 17s $\pm$ 5m |
| | MOLLEO (BIOT5) | 1h 23m 6s $\pm$ 20m |
| | MOLLEO (GPT-4) | 4h 06m 12s $\pm$ 1h 15min |

In this table, we show that for lightweight oracles, incorporating LLM edits indeed results in high runtimes since calling the LLM is more expensive than random edits. However, as experiments become more expensive, such as with docking, the cost of the LLM call becomes insignificant in comparison to the docking time, hence the runtime is governed by the oracle call. For MOLLEO (GPT-4), the runtime is further constrained by OpenAI API rate limits, which impose restrictions on the number of input/output tokens processed per minute.

### D.5 SIMILARITY ANALYSIS FOR MOLECULES BEFORE AND AFTER LLM EDITING

To investigate the types of LLM edits occurring to a molecule, we analyzed the Tanimoto similarity between molecules before and after editing, comparing these values to the similarity with a random molecule. This approach allows us to evaluate whether the edited molecules are as distant from their original versions as they are from random molecules.

We present these similarities in Table A13 for all the LLMs included in our study. For GPT-4, which employs crossover instead of mutations, we report both the maximum and minimum Tanimoto similarities, where the maximum similarity corresponds to the closer parent and the minimum similarity to the further parent. In all cases, the Tanimoto similarity between molecules before and after editing is consistently higher than the similarity between edited molecules and random ones, indicating that the edits effectively preserve molecular substructures.

Interestingly, molecules edited by BioT5 exhibit lower similarity to their pre-edit versions compared to other LLMs, suggesting that BioT5 may reconstruct more of the sequences.

Table A13: Tanimoto similarity between molecules after LLM editing and molecules before LLM editing/random sampled molecules

| | Tanimoto similarity to molecules before editing | Tanimoto similarity to random sampled molecules |
|---|---|---|
| MoleculeSTM | $0.761 \pm 0.235$ | $0.120 \pm 0.044$ |
| BioT5 | $0.173 \pm 0.082$ | $0.111 \pm 0.038$ |
| GPT-4 | $0.433 \pm 0.207$ (max) $0.165 \pm 0.089$ (min) | $0.123 \pm 0.045$ |

Table A14: Task keyword frequency in open-source LLM training data. We note that BioT5 has two training phases (pre-training and fine-tuning), and so we include the frequency of keywords in both datasets.

| Prompt | Keyword | MolSTM hits | BioT5 hits (Pre-training+Fine-tuning) |
|---|---|---|---|
| "This molecule inhibits JNK3" | JNK3 | 0 | 0+0 |
| | kinase | 1698 | 1698+271 |
| | Jun N-terminal kinase | 17 | 17+0 |
| "This molecule inhibits DRD3" | DRD3 | 0 | 0+0 |
| | Dopamine receptor | 73 | 73+3 |
| | Dopamine receptor d | 7 | 7+0 |
| | Dopamine receptor d3 | 0 | 0 |
| "This molecule inhibits EGFR" | EGFR | 82 | 82+0 |
| "This molecule binds to adenosine receptor A2a" | adenosine receptor A2a | 0 | 0+0 |
| | adenosine receptor | 25 | 25+3 |

Table A15: Multi objective results. The best model for each task is bolded.

| Task 1: maximize QED ($\uparrow$), minimize SA ($\downarrow$), maximize JNK3 ($\uparrow$) | | Summation (Top-10 AUC) ($\uparrow$) | Hypervolume ($\uparrow$) | Structural diversity ($\uparrow$) | Objective diversity ($\uparrow$) |
|---|---|---|---|---|---|
| Summation | Graph-GA | 1.967 ± 0.088 | 0.713 ± 0.083 | 0.741 ± 0.115 | 0.351 ± 0.079 |
| | MOLLEO (MOLSTM) | 2.177 ± 0.178 | 0.625 ± 0.162 | 0.803 ± 0.011 | **0.362 ± 0.074** |
| | MOLLEO (BIOT5) | 1.946 ± 0.222 | 0.592 ± 0.199 | **0.805 ± 0.196** | 0.341 ± 0.091 |
| | MOLLEO (GPT-4) | 2.367 ± 0.044 | **0.752 ± 0.085** | 0.726 ± 0.063 | 0.292 ± 0.076 |
| Pareto optimality | Graph-GA | 2.120 ± 0.159 | 0.603 ± 0.082 | 0.761 ± 0.034 | 0.219 ± 0.117 |
| | MOLLEO (MOLSTM) | 2.234 ± 0.246 | 0.472 ± 0.248 | 0.739 ± 0.015 | 0.306 ± 0.085 |
| | MOLLEO (BIOT5) | 2.325 ± 0.164 | 0.630 ± 0.120 | 0.724 ± 0.020 | 0.339 ± 0.062 |
| | MOLLEO (GPT-4) | **2.482 ± 0.057** | 0.727 ± 0.038 | 0.745 ± 0.057 | 0.322 ± 0.104 |
| Task 2: maximize QED ($\uparrow$), minimize SA ($\downarrow$), maximize GSKB3 ($\uparrow$) | | | | | |
| Summation | Graph-GA | 2.186 ± 0.069 | 0.719 ± 0.055 | 0.778 ± 0.122 | 0.379 ± 0.101 |
| | MOLLEO (MOLSTM) | 2.349 ± 0.132 | 0.303 ± 0.024 | **0.820 ± 0.010** | **0.440 ± 0.037** |
| | MOLLEO (BIOT5) | 2.306 ± 0.120 | 0.693 ± 0.093 | 0.803 ± 0.013 | 0.384 ± 0.045 |
| | MOLLEO (GPT-4) | 2.543 ± 0.014 | **0.832 ± 0.024** | 0.715 ± 0.052 | 0.391 ± 0.021 |
| Pareto optimality | Graph-GA | 2.339 ± 0.139 | 0.640 ± 0.034 | 0.816 ± 0.028 | 0.381 ± 0.071 |
| | MOLLEO (MOLSTM) | 2.340 ± 0.254 | 0.202 ± 0.054 | 0.770 ± 0.017 | 0.188 ± 0.010 |
| | MOLLEO (BIOT5) | 2.299 ± 0.203 | 0.645 ± 0.127 | 0.759 ± 0.022 | 0.371 ± 0.047 |
| | MOLLEO (GPT-4) | **2.631 ± 0.023** | 0.820 ± 0.024 | 0.646 ± 0.017 | 0.191 ± 0.026 |
| Task 3: maximize QED ($\uparrow$), JNK3 ($\uparrow$), minimize SA ($\downarrow$), GSKB3 ($\downarrow$), DRD2 ($\downarrow$) | | | | | |
| Summation | Graph GA | 3.856 ± 0.075 | 0.162 ± 0.048 | 0.821 ± 0.024 | 0.226 ± 0.057 |
| | MOLLEO (MOLSTM) | 4.040 ± 0.097 | 0.474 ± 0.193 | 0.783 ± 0.027 | **0.413 ± 0.064** |
| | MOLLEO (BIOT5) | 3.904 ± 0.092 | 0.266 ± 0.201 | **0.828 ± 0.005** | 0.243 ± 0.081 |
| | MOLLEO (GPT-4) | 4.017 ± 0.048 | 0.606 ± 0.086 | 0.726 ± 0.064 | 0.289 ± 0.050 |
| Pareto optimality | Graph GA | 4.051 ± 0.155 | 0.606 ± 0.052 | 0.688 ± 0.047 | 0.294 ± 0.074 |
| | MOLLEO (MOLSTM) | 3.989 ± 0.145 | 0.381 ± 0.204 | 0.792 ± 0.030 | 0.258 ± 0.019 |
| | MOLLEO (BIOT5) | 3.946 ± 0.115 | 0.367 ± 0.177 | 0.784 ± 0.020 | 0.367 ± 0.177 |
| | MOLLEO (GPT-4) | **4.212 ± 0.034** | **0.696 ± 0.029** | 0.641 ± 0.037 | 0.266 ± 0.062 |

## D.6 CAPTION ANALYSIS FOR OPEN-SOURCE LLMS

To understand how well objective tasks are represented in the training data of the underlying open-source LLMs used in this study, we examine the frequency of keywords related to those tasks in Table A14. Based on our findings, the models have seen a small number of data points related to molecules on objective tasks such as EGFR and Jun N-terminal kinase (although the latter uses a different spelling than we indicated with our prompt). To our knowledge, the LLMs do not contain information explicitly on tasks such as DRD3 and Adenosine receptor A2a, but related concepts do exist in the training data. Despite this, MOLLEO (BIOT5) achieves the best performance on these docking experiments (see Figure 3). These results highlight the strong generalization capabilities of LLMs.

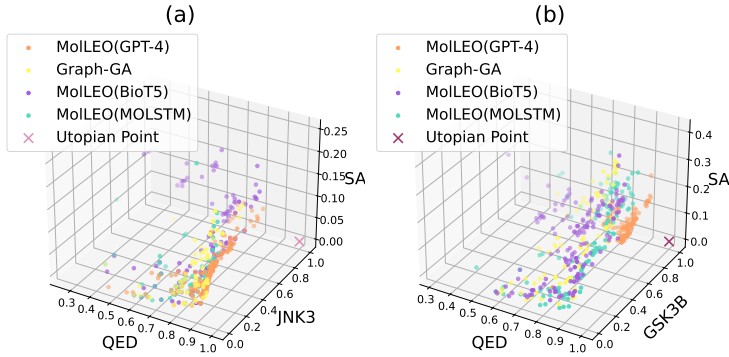

Figure A3: Pareto frontier visualizations for Graph-GA and MOLLEO on the following multi-objective tasks: (a) Task 1 (min SAscore, max JNK3 binding, max QED) and (b) Task 2 (min SAscore, max GSK3$\beta$ binding, max QED). The utopian point corresponds to the maximum (best) possible values across all objectives. SA scores are rescaled to $[0, 1]$.

## D.7 DIVERSITY ANALYSIS IN MULTI-OBJECTIVE OPTIMIZATION

We show the structural diversity and objective diversity for multi-objective optimization in Table A15. Structural diversity reflects the chemical diversity of the Pareto set and is computed by taking the average pairwise Tanimoto distance between Morgan fingerprints of molecules in the set. Objective diversity illustrates the objective value coverage of the Pareto frontier and is computed by taking the pairwise Euclidean distance between objective values of molecules in the Pareto set. In Figure A3, we also visualize the Pareto optimal set (in objective space) for MOLLEO and Graph-GA for Tasks 1 and 2.

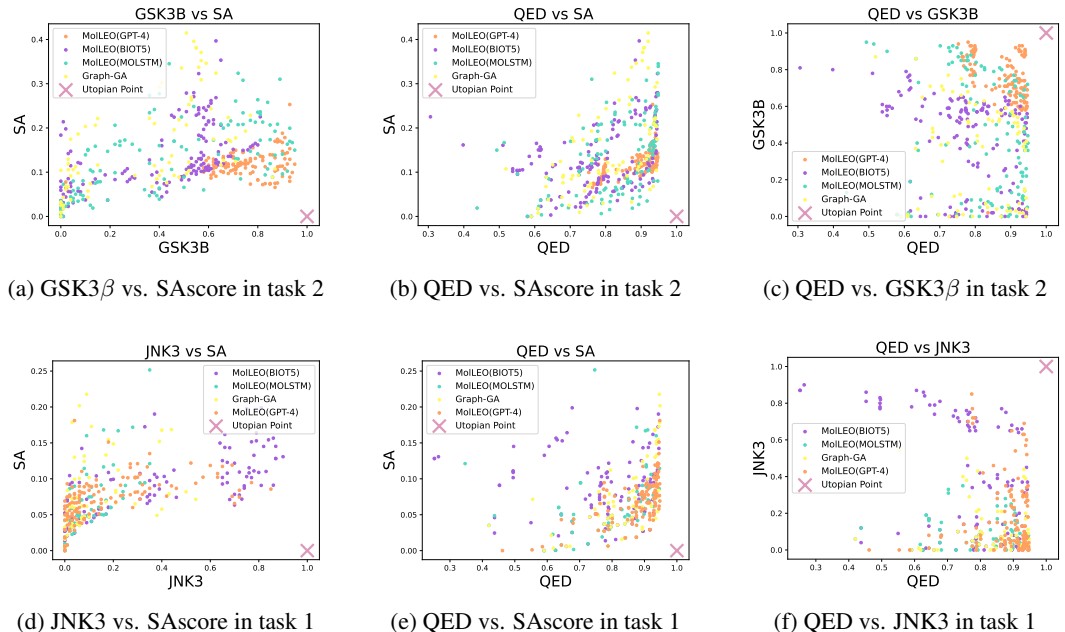

(a) GSK3$\beta$ vs. SAscore in task 2    (b) QED vs. SAscore in task 2    (c) QED vs. GSK3$\beta$ in task 2

(d) JNK3 vs. SAscore in task 1    (e) QED vs. SAscore in task 1    (f) QED vs. JNK3 in task 1

Figure A4: 2D plots for multi-objective optimization in task 1 and task 2

## D.8 CASE STUDY: SAMPLE MOLECULES FROM FINAL POOL

Below, we show the top ten molecules across all runs from the MOLLEO and Graph-GA for two tasks: `deco_hop` and `EGFR docking`.

### D.8.1 TASK 1: deco_hop

The goal of the deco_hop task is to generate molecules that contain specific substructures while not containing others; these substructures are shown in Figure A5. The final deco_hop score is calculated as the mean of substructure presence/absence (binary score) and Tanimoto distance to the target molecule. We showcase our best-generated molecules from the deco_hop task in Figure A6.

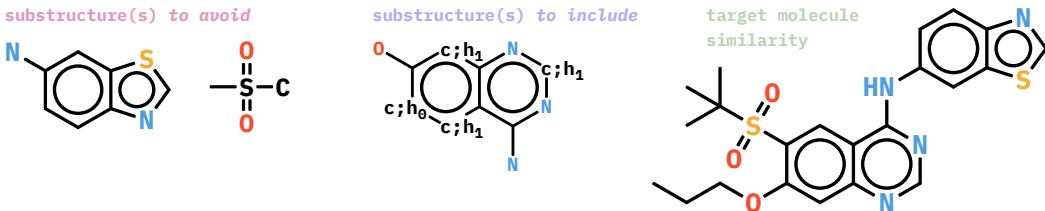

Figure A5: **Substructures to be included or avoided in the deco_hop task.**

### D.8.2 TASK 2: EGFR docking

The goal of the EGFR docking task is to generate molecules that have a low binding affinity to epidermal growth factor receptors in humans (EGFR, PBD ID: 2RGP. Molecules are docked against EGFR using AutoDock Vina (Eberhardt et al., 2021), and the output is the docking score of the binding process. We showcase our best-generated molecules from this task in Figure A7.

## E PROMPTS

For each model, we show the prompts used for each task. When creating the prompts, we followed the format of examples in the original source code as closely as possible.

---

**MOLLEO (MOLSTM) prompts**

**QED**
This molecule is like a drug.

**JNK3**
This molecule inhibits JNK3.

**GSK3$\beta$**
This molecule inhibits GSK3B.

**DRD2**
This molecule inhibits DRD2.

**mestranol_similarity**
This molecule looks like Mestranol.

**albuterol_similarity**
This molecule looks like Albuterol.

**thiothixene_rediscovery**
This molecule looks like Thiothixene.

**celecoxib_rediscovery**
This molecule looks like Celecoxib.

**perindopril_mpo**
This molecule looks like Perindopril and has 2 aromatic rings.

**ranolazine_mpo**
This molecule looks like Ranolazine, is highly permeable, is hydrophobic, and has 1 F atom.

**sitagliptin_mpo**
This molecule has the formula C16H15F6N5O, looks like Sitagliptin, is highly permeable, and is hydrophobic.

---

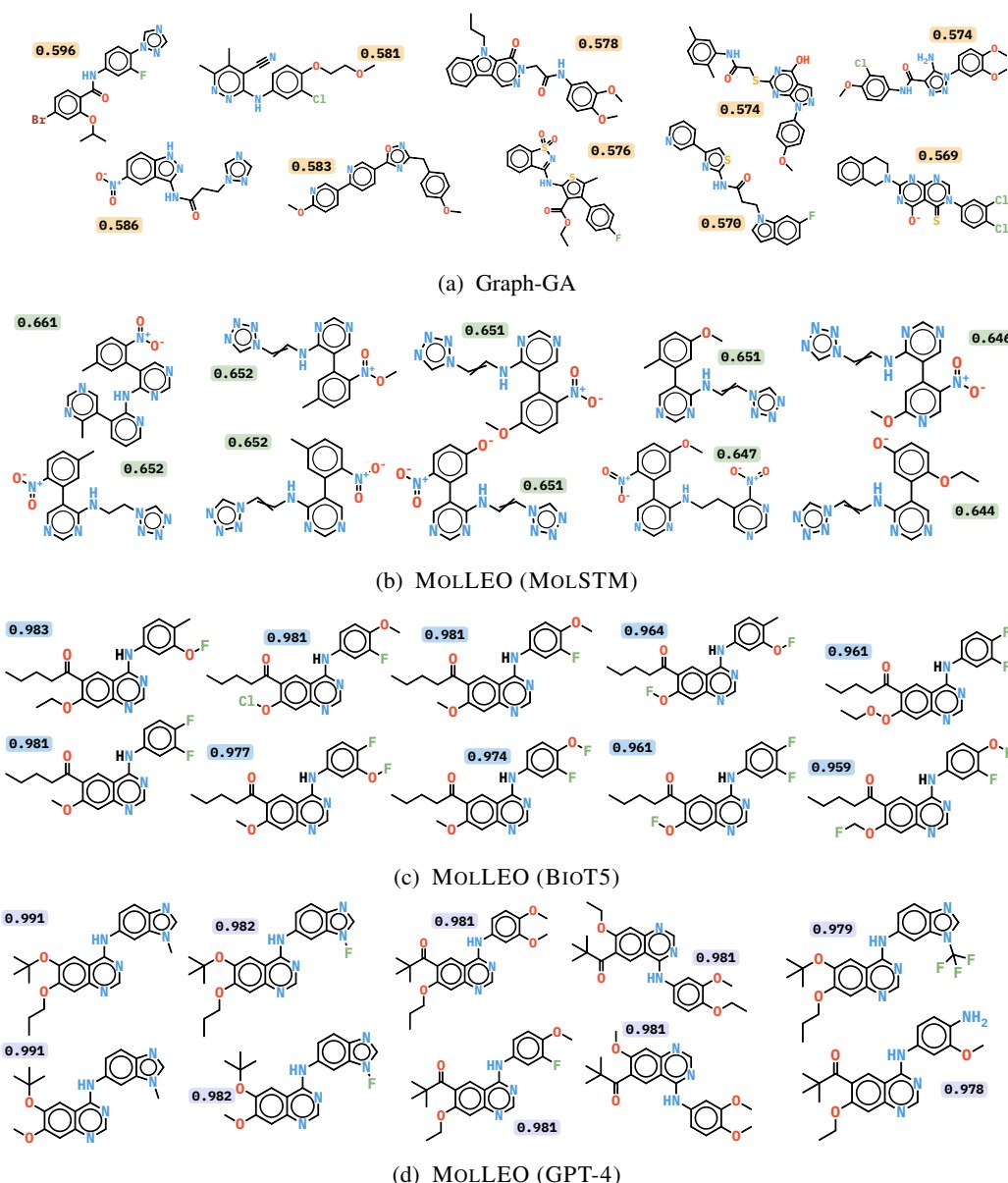

Figure A6: Molecules with best deco_hop scores generated by Graph-GA and each MoLLEO model. The deco_hop score of each molecule is written beside it. Higher deco_hop scores are better.

```
Isomers_C9H10N2O2PF2Cl
This molecule has the atoms C9H10N2O2PF2Cl.

deco_hop
This molecule does not contain the substructure [#7]-c1ccc2ncsc2c1, which is a
6-aminobenzothiazole, does not contain the substructure CS([#6])(=O)=O, which is a
dimethyl sulfone, contains the scaffold, which is a 4-amino-7-hydroxyquinazoline, and
is similar to CCCOc1cc2ncnc(Nc3ccc4ncsc4c3)c2cc1S(=O)(=O)C(C)(C)C.

scaffold_hop
This molecule does not contain the scaffold [#7]-c1n[c;h1]nc2[c;h1]c(-[#8])[c;h0]
[c;h1]c12, contains the substructure [#6]-[#6]-[#6]-[#8]-[#6]~[#6]~[#6]~[#6]~[#6]-
[#7]-c1ccc2ncsc2c1, and is similar to CCCOc1cc2ncnc(Nc3ccc4ncsc4c3)c2cc1S(=O)(=O)C(C)(C)C.

maxjnk3_maxqed_minsa
This molecule is synthesizable, looks like a drug, and inhibits JNK3.
```

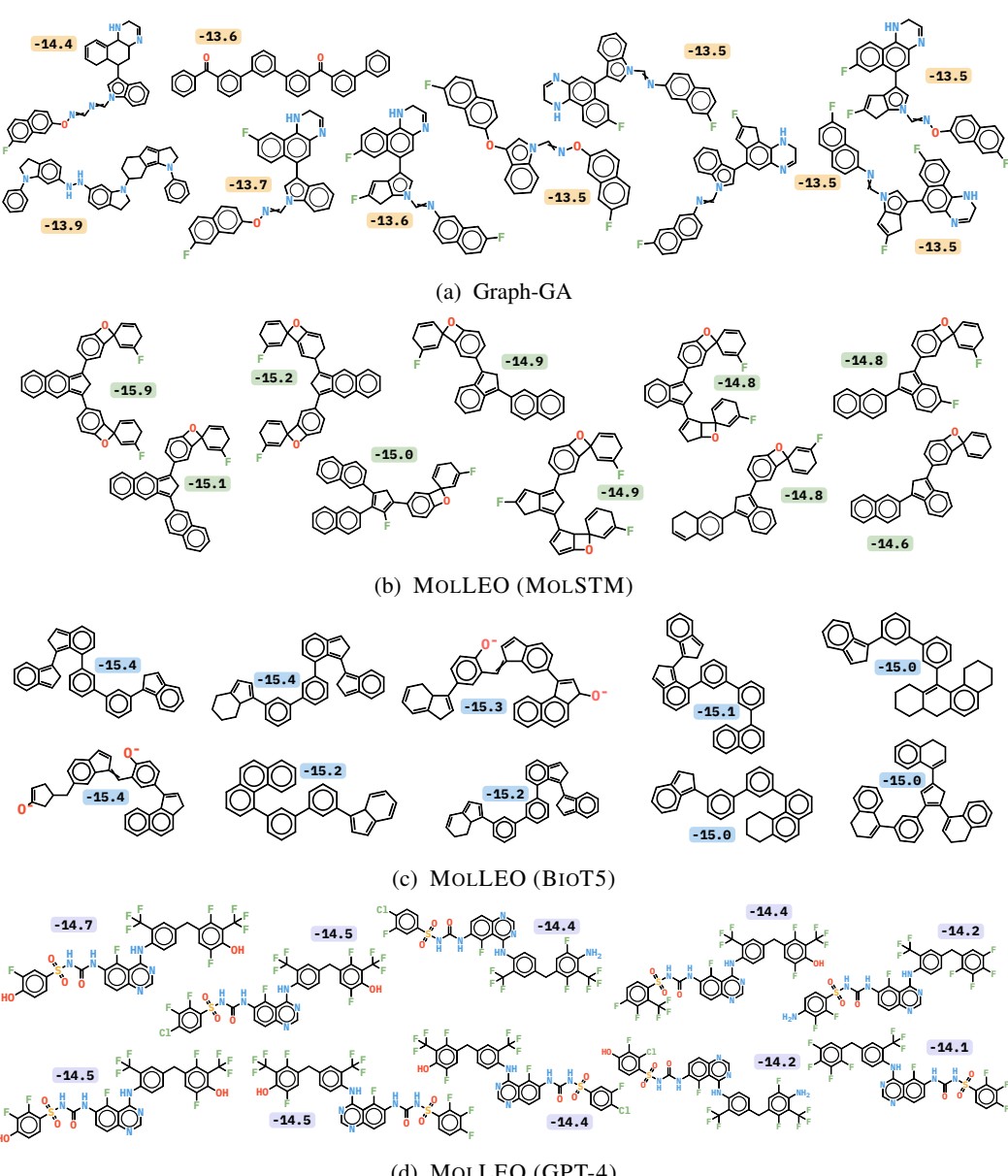

Figure A7: Molecules with best EGFR docking scores generated by Graph-GA and each MOLLEO model. The docking score of each molecule is written beside it. Lower docking scores are better.

```
maxgsk3b_maxqed_minsa
This molecule is synthesizable, looks like a drug, and inhibits GSK3B.

maxjnk3_maxqed_minsa_mindrd2_mingsk3b
This molecule is synthesizable, does not inhibit GSKB3, does not inhibit DRD2, looks
like a drug, and inhibits JNK3.

3pbl_docking
This molecule inhibits DRD3.

2rgp_docking
This molecule inhibits EGFR.

3eml_docking
This molecule binds to adenosine receptor A2a.
```

---

**MOLLEO (BIOT5) prompts**

*Template:*

Definition: You are given a molecule SELFIES. Your job is to generate a
SELFIES molecule that {OBJECTIVE}. Now complete the following example - Input:
<bom>{selfies_input}<eom> Output:

**QED**
OBJECTIVE: looks more like a drug

**JNK3**
OBJECTIVE: inhibits JNK3 more

**GSK3$\beta$**
OBJECTIVE: inhibits GSK3B more

**DRD2**
OBJECTIVE: inhibits DRD2 more

**mestranol_similarity**
OBJECTIVE: looks more like Mestranol

**albuterol_similarity**
OBJECTIVE: looks more like Albuterol.

**thiothixene_rediscovery**
OBJECTIVE: looks more like Thiothixene

**celecoxib_rediscovery**
OBJECTIVE: looks more like Celecoxib.

**perindopril_mpo**
OBJECTIVE: looks more like Perindopril and has 2 aromatic rings

**sitagliptin_mpo**
OBJECTIVE: has the formula C16H15F6N5O, looks more like Sitagliptin, is highly
permeable, and is hydrophobic

**ranolazine_mpo**
OBJECTIVE: looks more like Ranolazine, is highly permeable, is hydrophobic, and has 1 F
atom

**Isomers_C9H10N2O2PF2Cl**
OBJECTIVE: has the formula C9H10N2O2PF2Cl

**deco_hop**
OBJECTIVE: does not contain the substructure [#7]-c1ccc2ncsc2c1, does
not contain the substructure CS([#6])(=O)=O, contains the scaffold
[#7]-c1n[c;h1]nc2[c;h1]c(-[#8])[c;h0][c;h1]c12, and is similar to
[C][C][C][O][C][=C][C][=N][C][=N][C][Branch1][#C][N][C][=C][C][=C][N][=C][S][C][Ring1]
[Branch1][=C] [Ring1][=Branch2][=C][Ring1][S][C][=C][Ring2][Ring1][Ring2][S][=Branch1]
[C][=O][=Branch1][C][=O][C][Branch1][C][C][Branch1][C][C][C]

**scaffold_hop**
OBJECTIVE: does not contain the scaffold [#7]-c1n[c;h1]nc2[c;h1]c(-[#8])[c;h0][c;h1]c12,
contains the substructure [#6]-[#6]-[#6]-[#8]-[#6]~[#6]~[#6]~[#6]~[#6]-
[#7]-c1ccc2ncsc2c1, and is similar to the SELFIES [C][C][C][O][C][=C][C][=N][C][=N][C]
[Branch1][#C][N][C][=C][C][=C][N][=C][S] [C][Ring1][Branch1][=C][Ring1][=Branch2][=C]
[Ring1][S][C][=C][Ring2][Ring1][Ring2][S] [=Branch1][C][=O][=Branch1]
[C][=O][C][Branch1][C][C][Branch1] [C][C][C]

**maxjnk3_maxqed_minsa**
OBJECTIVE: is a greater inhibitor of JNK3, is more synthesizable and is more like a
drug.

**maxgsk3b_maxqed_minsa**
OBJECTIVE: inhibits GSK3B more, is more synthesizable and is more like a drug.

**maxjnk3_maxqed_minsa_mindrd2_mingsk3b**
OBJECTIVE: is a greater inhibitor of JNK3, is more like a drug, inhibits GSK3B less,
inhibits DRD2 less and is more synthesizable.

**3pbl_docking**
OBJECTIVE: inhibits DRD3 more

**2rgp_docking**
OBJECTIVE: inhibits EGFR more

---

**3eml_docking**
OBJECTIVE: binds better to adenosine receptor A2a

## MOLLEO (GPT-4) prompts

*Template:*

I have two molecules and their {TASK}. {OBJECTIVE_DEFINITION}

(Smiles of Parent A, objective score of Parent A) (Smiles of Parent B, objective score of Parent B)

Please propose a new molecule that {OBJECTIVE}. You can either make crossover and mutations based on the given molecules or just propose a new molecule based on your knowledge.
Your output should follow the format: {«<Explanation»>: $EXPLANATION, «<Molecule»>: box{$Molecule}}. Here are the requirements:
1. $EXPLANATION should be your analysis.
2. The $Molecule should be the smiles of your proposed molecule.
3. The molecule should be valid.

**QED:**
OBJECTIVE: has a higher QED score
TASK: QED scores
OBJECTIVE_DEFINITION: The QED score measures the drug-likeness of the molecule.

**JNK3**
OBJECTIVE: has a higher JNK3 score
TASK: JNK3 scores
OBJECTIVE_DEFINITION: The JNK3 score measures a molecular's biological activity against JNK3.

**GSK3$\beta$**
OBJECTIVE: has a higher GSK3$\beta$ score
TASK: GSK3$\beta$ scores
OBJECTIVE_DEFINITION: The GSK3$\beta$ score measures a molecular's biological activity against GSK3$\beta$.

**DRD2**
OBJECTIVE: has a higher DRD2 score
TASK: DRD2 scores
OBJECTIVE_DEFINITION: The DRD2 score measures a molecule's biological activity against a biological target named the dopamine type 2 receptor (DRD2).

**mestranol_similarity**
OBJECTIVE: has a higher mestranol similarity score
TASK: mestranol similarity scores
OBJECTIVE_DEFINITION: The mestranol similarity score measures a molecule's Tanimoto similarity with Mestranol.

**thiothixene_rediscovery**
OBJECTIVE: has a higher thiothixene rediscovery score
TASK: thiothixene rediscovery scores
OBJECTIVE_DEFINITION: The thiothixene rediscovery score measures a molecule's Tanimoto similarity with thiothixene's SMILES to check whether it could be rediscovered.

**perindopril_mpo**
OBJECTIVE: has a higher perindopril multi-objective score
TASK: perindopril multi-objective scores
OBJECTIVE_DEFINITION: The perindopril multi-objective score measures the geometric means of several scores, including the molecule's Tanimoto similarity to perindopril and the number of aromatic rings.

**sitagliptin_mpo**
OBJECTIVE: has a higher sitagliptin multi-objective score
TASK: sitagliptin multi-objective scores
OBJECTIVE_DEFINITION: The sitagliptin multi-objective score measures the geometric means of several scores, including the molecule's Tanimoto similarity to sitagliptin, TPSA score, LogP score and isomer score with C16H15F6N5O.

**ranolazine_mpo**
OBJECTIVE: has a higher ranolazine multi-objective score
TASK: ranolazine multi-objective scores
OBJECTIVE_DEFINITION: The ranolazine multi-objective score measures the geometric means of several scores, including the molecule's Tanimoto similarity to ranolazine, TPSA score LogP score and number of fluorine atoms.

```
Isomers_C9H10N2O2PF2Cl:
OBJECTIVE: has a higher isomer score
TASK: isomer scores
OBJECTIVE_DEFINITION: The isomer score measures a molecule's similarity in terms of
atom counter to C9H10N2O2PF2Cl.

deco_hop
OBJECTIVE: has a higher deco hop score
TASK: deco hop scores
OBJECTIVE_DEFINITION: The deco hop score is the arithmetic means of several scores,
including binary score about whether contain certain SMARTS structures (maximize
the similarity to the SMILE '[#7]-c1n[c;h1]nc2[c;h1]c(-[#8])[c;h0][c;h1]c12',
while excluding specific SMARTS patterns '[#7]-c1ccc2ncsc2c1' and
'CS([#6])(=O)=O') and (2) the molecule's Tanimoto similarity to PHCO
'CCCOc1cc2ncnc(Nc3ccc4ncsc4c3)c2cc1S(=O)(=O)C(C)(C)C'.

scaffold_hop
OBJECTIVE: has a higher scaffold hop score
TASK: scaffold hop scores
OBJECTIVE_DEFINITION: The scaffold hop score is the arithmetic means
of several scores, including (1) binary score about whether contains
certain SMARTS structures (maximize the similarity to the SMILE
'[#6]-[#6]-[#6]-[#8]-[#6]~[#6]~[#6]~[#6]~[#6]-[#7]-c1ccc2ncsc2c1', while excluding
specific SMARTS patterns '[#7]-c1n[c;h1]nc2[c;h1]c(-[#8])[c;h0][c;h1]c12') and
(2) the molecule's Tanimoto similarity to PHCO 'CCCOc1cc2ncnc(Nc3ccc4ncsc4c3)c2cc1S
(=O)(=O)C(C)(C)C'.

maxjnk3_maxqed_minsa
OBJECTIVE: has a higher QED score, a higher JNK3 score, and a lower SA score
TASK: QED, SA (Synthetic Accessibility), and JNK3 scores.
OBJECTIVE_DEFINITION: None

maxgsk3b_maxqed_minsa
OBJECTIVE: has a higher QED score, a higher GSK3β score, and a lower SA score
TASK: QED, SA (Synthetic Accessibility), and GSK3β scores
OBJECTIVE_DEFINITION: None

maxjnk3_maxqed_minsa_mindrd2_mingsk3b
OBJECTIVE: has a higher QED score, a higher JNK3 score, a lower GSK3β score, a lower
DRD2 score and a lower SA score
TASK: QED, SA (Synthetic Accessibility), JNK3, GSK3β and DRD2 scores
OBJECTIVE_DEFINITION: None

2rgp_docking
OBJECTIVE: binds better to EGFR
TASK: docking scores to EGFR
OBJECTIVE_DEFINITION: The docking score measures how well a molecule binds to EGFR. A
lower docking score generally indicates a stronger or more favorable binding affinity.

3pbl_docking
OBJECTIVE: binds better to DRD3
TASK: docking scores to DRD3
OBJECTIVE_DEFINITION: The docking score measures how well a molecule binds to DRD3.  A
lower docking score generally indicates a stronger or more favorable binding affinity.

3eml_docking
OBJECTIVE: binds better to adenosine receptor A2a
TASK: docking scores to adenosine receptor A2a
OBJECTIVE_DEFINITION: The docking score measures how well a molecule binds to adenosine
receptor A2a.  A lower docking score generally indicates a stronger or more favorable
binding affinity.
```

