# OpenReview forum: "Efficient Evolutionary Search Over Chemical Space with Large Language Models"
_ICLR.cc/2025/Conference — ICLR 2025 Poster_

### Official Review · Reviewer_bF3X · 2024-10-29

**Soundness:** 3
**Presentation:** 3
**Contribution:** 2
**Rating:** 8
**Confidence:** 4

**Summary:**

This paper presents a novel approach to molecular optimization by replacing the crossover and mutation operations in GraphGA with Large Language Models (LLMs) to enhance sample efficiency. Instead of employing LLMs for end-to-end molecule generation, the study uniquely integrates LLM capabilities into specific genetic algorithm operations (crossover, mutation), effectively combining the strengths of LLMs with those of GAs. I really enjoyed reading this paper and think it is a good one. However, the method seems too simple, and I believe that if the methodology is reinforced, it could become an excellent paper.

**Strengths:**

1.	Unlike conventional end-to-end LLM methods, the paper strategically replaces only the GA operations where LLMs can be effective. This selective substitution leverages the advantages of LLM and genetic algorithm.
2.	It proposes a prompting method that utilizes LLMs to improve performance in molecular optimization tasks. This approach opens avenues for further advancements if combined with additional methods.

**Weaknesses:**

1.	While the main contribution appears to be the replacement of evolutionary algorithm (EA) components, the paper lacks a comprehensive analysis of existing EA methods. Particularly in multi-objective optimization, evolutionary algorithms like MOEA/D and NSGA-III [1, 2]. The study introduces Pareto Optimality (PO) using the Pareto Front but does not compare it with these established multi-objective optimization EAs. Additionally, the paper DyMol [3] has proposed a Pareto Sampling method similar to the PO in this study, which should be acknowledged.

2.	The MolLEO framework is applied only to GraphGA, which limits its demonstrated versatility. Recent works like Genetic GFN [4] and Saturn [5] have enhanced performance by incorporating crossover and mutation into methods like Augmented Memory and GRU-based GFlowNet. Applying MolLEO to these methods could further validate its effectiveness.

3.	The paper could better articulate why specific objectives were chosen, such as maximizing JNK3 and minimizing GSK3β in objective selectivity tasks. Providing rationale for these choices would strengthen the study's context and applicability.

[1] Zhang et. al. (2007) "MOEA/D: A multiobjective evolutionary algorithm based on decomposition." IEEE Transactions on evolutionary computation

[2] Deb and Jain, (2013) "An evolutionary many-objective optimization algorithm using reference-point-based nondominated sorting approach, part i: solving problems with box constraints." IEEE transactions on evolutionary computation

[3] Shin et. al. (2024) "DyMol: Dynamic Many-Objective Molecular Optimization with Objective Decomposition and Progressive Optimization." ICLR 2024 Workshop on Generative and Experimental Perspectives for Biomolecular Design.

[4] Kim et. al. (2024) "Genetic-guided GFlowNets: Advancing in Practical Molecular Optimization Benchmark." Neurips.

[5] Guo, and Philippe, (2024) "Saturn: Sample-efficient Generative Molecular Design using Memory Manipulation." arXiv preprint arXiv:2405.17066.

**Questions:**

1.	In replacing the crossover and mutation operations within GraphGA, did the LLM genuinely perform these operations, or did it retrieve known SMILES strings based on the provided inputs? For mutation, does the LLM modify only certain tokens in the existing SMILES, similar to the actual mutation operation, or does it reconstruct the entire sequence? Regarding crossover, how closely does the LLM's output resemble true crossover? Given the prompt used—You can either make crossover and mutations based on the given molecules or “just propose a new molecule based on your knowledge”—it's important to understand the ratio of outputs resulting from crossover, mutation, and LLM retrieval.
2.	The objectives in the PMO benchmark are well-known objectives, with extensive experimental data available. Consequently, the LLM might already be familiar with molecules that perform well for these objectives, especially for targets like JNK3 mentioned in the introduction. Is there a possibility that the LLM is retrieving and outputting known high-performing molecules rather than generating novel ones? Verifying this would be crucial to assess the true innovative capacity of the approach.
3.	Building on the previous question, did you intentionally leverage the LLM's existing knowledge to rapidly improve performance, perhaps aiming for a zero-shot learning scenario? If so, this strategy has merit. However, it raises concerns about the method's applicability to entirely new objectives where the LLM lacks prior knowledge. How do you address this potential limitation?
4.	As highlighted in Weakness 2, the evaluation of MolLEO is confined to GraphGA. To demonstrate its generalizability and robustness, it would be beneficial to apply the MolLEO framework to other models like Genetic GFN and Saturn.
5.	All the score curves in the paper compare GraphGA and MolLEO. Including Augmented Memory in these comparisons would provide a more comprehensive evaluation of MolLEO's performance relative to other established methods.
6.	In Table 2, the "Sum" appears to represent the sum of AUC top-10 scores. To prevent confusion—since row “sum“ and column “sum“ are denoted same—please clarify this in the caption or relabel it as "Sum(AUC)." Alternatively, consider using a term like "Scalarization" to more accurately reflect the data presented.
7.	Since the objectives involve both maximization and minimization, summing their scores may not straightforwardly reflect overall performance. Maybe you applied simple (e.g., using 1−score) score for minimization objectives for scalarization. It's important to explicitly detail these methods to ensure the validity and reproducibility of your results.

---

> ### Author Response · Authors · 2024-11-22
>
> We appreciate the reviewer’s constructive feedback and questions. We are glad that the reviewer considers our method to be “strategic” and “effective”, leveraging the advantages of both LLMs and GAs. We wholeheartedly agree with the reviewer that this method can “open avenues for further advancements”.
>
> We focus on increasing the breadth of our experiments to new GA settings, as well as additional analyses requested by the reviewer. We have updated our PDF in blue text to highlight these changes and address them in detail below.
>
> > While the main contribution appears to be the replacement of evolutionary algorithm (EA) components, the paper lacks a comprehensive analysis of existing EA methods. Particularly in multi-objective optimization, evolutionary algorithms like MOEA/D and NSGA-III [1, 2]. The study introduces Pareto Optimality (PO) using the Pareto Front but does not compare it with these established multi-objective optimization EAs. Additionally, the paper DyMol [3] has proposed a Pareto Sampling method similar to the PO in this study, which should be acknowledged.
>
> We thank the reviewer for bringing these works to our attention. We agree that there are numerous intriguing multi-objective optimization algorithms worth exploring. However, as the reviewer rightly noted, this is not the focus of our contributions in this paper. Here, we have chosen to employ the simplest widely-used multi-objective optimization methods commonly used in practice (PO and summation). To our understanding, PO differs from [1,2] by simply keeping all the solutions in the Pareto set, while the other methods incorporate some selection criteria of clustering for selecting the set. That said, the works mentioned [1,2,3] by the reviewer are indeed compelling; inspired by this, we have added a new related works section on multi-objective algorithms (Appendix A1) where we discuss all these works, as well as some additional ones we found. We believe these approaches would be valuable to investigate in future research.
>
> > The MolLEO framework is applied only to GraphGA, which limits its demonstrated versatility. Recent works like Genetic GFN [4] and Saturn [5] have enhanced performance by incorporating crossover and mutation into methods like Augmented Memory and GRU-based GFlowNet. Applying MolLEO to these methods could further validate its effectiveness.
>
> We thank the reviewer for this valuable suggestion. We selected GraphGA as the backbone model because its simplicity as a baseline allows us to isolate and directly evaluate the effects of the LLM-based genetic operators in our study. .
>
> Additionally, we have incorporated LLM-based genetic operators in Augmented Memory and integrated them into other EAs including Genetic GFN [4] and JANUS [6] as suggested by you and Reviewer TwFV. The results are shown below, as well as in  Appendix D.2 and Appendix D.3 of our updated manuscript.
>
> | Augmented Memory \\ Objective (↑) | w/o LLM       | w/ BioT5       | w/ GPT-4         |
> |--------------------------------|---------------|----------------|------------------|
> | JNK3                           | 0.773 ± 0.073 | 0.781 ± 0.094  | **0.794 ± 0.087** |
> | albuterol_similarity           | 0.918 ± 0.026 | 0.925 ± 0.076  | **0.941 ± 0.033** |
>
>
> | Genetic GFN \\ Objective (↑)       | w/o LLM       | w/ BioT5   | w/ GPT-4      |
> |-----------------------------|------------------|-------------------|---------------------|
> | JNK3                        | 0.766 ± 0.077    | 0.775 ± 0.056     | **0.783 ± 0.034**   |
> | Albuterol Similarity        | 0.946 ± 0.013    | 0.962 ± 0.017     | **0.971 ± 0.020**   |
>
>
>  | JANUS \\ Objective (↑) | w/o LLM       | w/ MolSTM       | w/ BioT5         |
> |--------------------------------|---------------|----------------|------------------|
> | JNK3                           | 0.678 ± 0.031 | 0.680 ± 0.024  | **0.685 ± 0.044** |
> | albuterol_similarity           | 0.712 ± 0.049 | 0.779 ± 0.038  | **0.904 ± 0.051** |
>
> We hope these results demonstrate the versatility of our proposed method.
>
> > The paper could better articulate why specific objectives were chosen, such as maximizing JNK3 and minimizing GSK3β in objective selectivity tasks. Providing rationale for these choices would strengthen the study's context and applicability.
>
> We appreciate the reviewer's insightful comment. JNK3 and GSK3β are widely used oracles for evaluating binding affinities. One valuable task in drug design is selectively designing molecular structures that exhibit high binding affinity for one target while minimizing affinity for another [7]. In our study, we chose to maximize JNK3 and minimize GSK3β as a possible selectivity design task, although the direction was chosen randomly.

---

> ### Author Response · Authors · 2024-11-22
>
> > In replacing the crossover and mutation operations within GraphGA, did the LLM genuinely perform these operations, or did it retrieve known SMILES strings based on the provided inputs? For mutation, does the LLM modify only certain tokens in the existing SMILES, similar to the actual mutation operation, or does it reconstruct the entire sequence? Regarding crossover, how closely does the LLM's output resemble true crossover?
>
> The reviewer raises excellent points here.  In general, LLMs are actually not that good at generating fit molecules from prompting alone (see Appendix C1, Table A3, where prompting LLMs to generate molecules de novo results in essentially a random pool). The question about what edits are actually taking place is a very interesting one – one way to answer this question is to analyze the Tanimoto similarity between molecules after and before editing, vs. with a random molecule and see if the edited molecules are as far away as the random ones, which is analogous to reconstructing the whole sequence. We show these similarities in the table below for all LLMs we used in the study. Because GPT-4 uses crossover instead of mutations, we report the max and min Tanimoto similarities, where max Tanimoto similarity is the closer parent and min Tanimoto similarity is the further parent. In all cases, the Tanimoto similarity between the molecules before and after editing are larger than after editing and with a random molecule, which strongly indicates that the edits are preserving molecular substructures. Interestingly, molecules edited by BioT5 exhibit lower similarity to their pre-edit versions compared to other LLMs, suggesting that BioT5 may reconstruct more of the sequences. We have also added this to Appendix D5 in the updated manuscript. We thank the reviewer again for this interesting suggestion!
>
> > The objectives in the PMO benchmark are well-known objectives, with extensive experimental data available. Is there a possibility that the LLM is retrieving and outputting known high-performing molecules rather than generating novel ones? Verifying this would be crucial to assess the true innovative capacity of the approach.
>
> We verify this for open-source LLMs. Specifically, we compared the final molecules generated for the JNK3 task against two comprehensive datasets: ZINC20, which was used to create the training set for BioT5, and PubChem, which served as the training set for MoleculeSTM. Our analysis revealed no overlap between the generated molecules and these datasets, indicating that the models do not retrieve and output known high-performing molecules (also – App. Table A3 shows that LLMs are not great at retrieval in the first place!). We also evaluated the framework's performance in a scenario resembling a discovery setting, where we initialized the optimization process with the best currently known JNK3 inhibitors (Table 3, main paper). Our method outperformed the baseline by identifying better new molecules, demonstrating its potential utility in novel discovery settings.
>
> > Building on the previous question, did you intentionally leverage the LLM's existing knowledge to rapidly improve performance, perhaps aiming for a zero-shot learning scenario? If so, this strategy has merit. However, it raises concerns about the method's applicability to entirely new objectives where the LLM lacks prior knowledge. How do you address this potential limitation?
>
> Yes, we intentionally use LLM’s existing knowledge to guide the search process in zero-shot settings without the need for finetuning. The reviewer raises an important point. Following the reviewer’s suggestion, we analyzed which protein-drug interaction tasks are represented in the training datasets of MoleculeSTM and BioT5, as detailed in Appendix Table A14. Notably, their training datasets do not explicitly include captions for two specific targets, DRD3 and Adenosine receptor A2a. However, they do contain information on related concepts, such as other adenosine and dopamine receptors. This suggests that knowledge from related tasks can be leveraged to optimize molecules for novel, unseen targets. A major strength of LLMs lies in their ability to generalize across diverse target functions. When optimizing for a genuinely novel task that the LLM has not encountered before, it is assumed that the model’s training on related tasks (e.g., optimizing for Jun N-kinase 3 while having been trained on other kinases) provides generalizable insights into the functional space of kinases. This transfer learning capability enables meaningful, functionally informed edits. In cases where the task is entirely novel—where no relevant or related information exists in the training data—the model may generate random molecules, akin to random mutation operations. Integrating LLMs into an optimization framework mitigates this limitation, as non-meaningful edits are filtered out.

---

> ### Author Response · Authors · 2024-11-22
>
> > All the score curves in the paper compare GraphGA and MolLEO. Including Augmented Memory in these comparisons would provide a more comprehensive evaluation of MolLEO's performance relative to other established methods.
>
> We agree with this very reasonable point; we have updated Figure A1 in the manuscript to include optimization curves for two additional top-performing methods: Augmented Memory (as requested by the reviewer) and REINVENT (a widely-used RL framework).
>
> > In Table 2, the "Sum" appears to represent the sum of AUC top-10 scores. To prevent confusion—since row “sum“ and column “sum“ are denoted same—please clarify this in the caption or relabel it as "Sum(AUC)." Alternatively, consider using a term like "Scalarization" to more accurately reflect the data presented.
>
> We thank the reviewer for this suggestion; this is much better. We have updated the caption in the manuscript.
>
> > Since the objectives involve both maximization and minimization, summing their scores may not straightforwardly reflect overall performance. Maybe you applied simple (e.g., using 1−score) score for minimization objectives for scalarization. It's important to explicitly detail these methods to ensure the validity and reproducibility of your results.
>
> The reviewer is correct–for the objectives involving minimization, we applied a simple transformation by using 1−score. Also, we ensure all objectives remain within the range of 0 to 1 by normalizing values to allow for comparability across objectives (except for docking scores; those are not scaled). We have updated these details in the manuscript (Appendix B).
>
> ### Closing comment
> We hope that our responses were sufficient in clarifying all the great questions asked by the reviewer. We sincerely thank the reviewer for their time and consideration, and we kindly encourage them to reevaluate their score if they find that our rebuttal and the new experiments presented warrant it.
>
> ### References
>
> [1] Zhang et. al. (2007) "MOEA/D: A multiobjective evolutionary algorithm based on decomposition." IEEE Transactions on evolutionary computation
>
> [2] Deb and Jain, (2013) "An evolutionary many-objective optimization algorithm using reference-point-based nondominated sorting approach, part i: solving problems with box constraints." IEEE transactions on evolutionary computation
>
> [3] Shin et. al. (2024) "DyMol: Dynamic Many-Objective Molecular Optimization with Objective Decomposition and Progressive Optimization." ICLR 2024 Workshop on Generative and Experimental Perspectives for Biomolecular Design.
>
> [4] Kim et. al. (2024) "Genetic-guided GFlowNets: Advancing in Practical Molecular Optimization Benchmark." Neurips.
>
> [5] Guo, and Philippe, (2024) "Saturn: Sample-efficient Generative Molecular Design using Memory Manipulation." arXiv preprint arXiv:2405.17066.
>
> [6] Nigam, AkshatKumar, Robert Pollice, and Alan Aspuru-Guzik. "JANUS: parallel tempered genetic algorithm guided by deep neural networks for inverse molecular design." arXiv preprint arXiv:2106.04011 (2021).
>
> [7] Schneuing, Arne, et al. "Structure-based drug design with equivariant diffusion models." arXiv preprint arXiv:2210.13695 (2022).

---

> > ### Author Response · Authors · 2024-11-25
> > **Looking Forward to Your Feedback**
> >
> > Dear Reviewer,
> >
> > We are very appreciative of the time you have spent reviewing our paper and providing very valuable feedback. As the rebuttal period is approaching the end, we wanted to see if the reviewer had any remaining concerns or questions. In our rebuttal, we have incorporated the reviewer’s excellent suggestions of comparing the differences between molecules before and after editing, as well as showing that our operations can improve the performance in three new backbone models. We have also analyzed the presence of task information in the training dataset to show that our method is not simply retrieving memorized content, and improved our discussion of multiobjective optimization methods.
> >
> > We would be happy to engage in further discussions with the reviewer regarding these topics or any additional concerns the reviewer has – please let us know! We thank the reviewer again for their time spent with our paper. If the reviewer finds that our new experiments, analyses, and discussions are useful for improving the paper, we would be grateful if the reviewer would potentially consider a fresher evaluation of our paper.

---

> > > ### Comment · Reviewer_bF3X · 2024-11-26
> > > **Official Comment by Reviewer bF3X**
> > >
> > > Thanks for answering my questions. The overall method has been enhanced, and all the issues I previously raised have been addressed. I am raising my score to 8. Great job!

---

> > > > ### Author Response · Authors · 2024-11-26
> > > >
> > > > We are thrilled! Thank you for your great feedback throughout the review process as well as your positive support.

---

> > > > ### Comment · Reviewer_bF3X · 2024-11-27
> > > >
> > > > Could you provide the code implementing augmented memory and genetic gflownet using MolLEO? It would be greatly appreciated.

---

> > > > > ### Author Response · Authors · 2024-11-28
> > > > >
> > > > > Dear reviewer,
> > > > >
> > > > > We provide our implementation of these two experiments at https://github.com/AnonymousSubmission-code-reproduce/rebuttal_exp/tree/main
> > > > >
> > > > > Please feel free to reach out with further questions.

---

### Official Review · Reviewer_TwFV · 2024-10-30

**Soundness:** 3
**Presentation:** 3
**Contribution:** 3
**Rating:** 8
**Confidence:** 4

**Summary:**

The paper introduces MOLLEO, an algorithm that combines chemistry-aware large language models (LLMs) with evolutionary algorithms to optimize molecular properties efficiently. The LLMs guide crossover and mutation operations, reducing evaluations needed in black-box optimization for molecular discovery.

**Strengths:**

MOLLEO showcases a novel application of LLMs within evolutionary processes, yielding superior performance across 26 molecular design tasks and showcasing promise in early drug discovery.

**Weaknesses:**

MOLLEO’s reliance on proprietary LLMs may limit reproducibility, and the diversity of the chemical space explored could be improved for broader molecular applications.

**Questions:**

Question 1: Could the authors provide a computational cost comparison between MOLLEO and traditional methods, given the LLM integration?

Question 2: How does MOLLEO balance chemical space exploration with optimization, and could additional mechanisms for diversity be beneficial?

---

> ### Author Response · Authors · 2024-11-22
>
> We thank the reviewer for their positive feedback! We are thrilled the reviewer finds our method to “yield superior performance” and “showcase promise in early drug discovery”.
>
> Below, we address the reviewer’s questions about computational costs as well as combining with other evolutionary mechanisms. We have updated our PDF in blue text to highlight these changes and address them in detail below.
>
>
> > Could the authors provide a computational cost comparison between MOLLEO and traditional methods, given the LLM integration?
>
> The most expensive part of chemistry experiments is often the oracle call. Below, we show the computational costs of MolLEO(BioT5) vs GraphGA for two experiments: JNK3, which uses a lightweight oracle, and docking against human dopamine D3 receptor (PDB ID 3pbl), which is more expensive.
> | Task               | Model      | Average task completion time (5 runs)     |
> |--------------------|------------|--------------|
> | **Docking 3pbl**           | Graph GA   | 8h 23m 37s   |
> |                    | MolLEO(BioT5)         | 6h 57m 14s   |
> |                    | MolLEO(GPT-4)        | 12h 14m 25s  |
> | **JNK3**   | Graph GA   | 12m 17s      |
> |                    | MolLEO(BioT5)         | 1h 23m 6s    |
> |                    | MolLEO(GPT-4)        | 4h 06m 12s   |
>
>
>
> In this table, we show that for lightweight oracles, incorporating LLM edits indeed results in high runtimes since calling the LLM is more expensive than random edits. However, as experiments become more expensive, such as with docking, the cost of the LLM call becomes insignificant in comparison to the docking time, hence the runtime is governed by the oracle call. For MolLEO (GPT-4), the runtime is further constrained by OpenAI API rate limits, which impose restrictions on the number of input/output tokens processed per minute.
>
> > How does MOLLEO balance chemical space exploration with optimization, and could additional mechanisms for diversity be beneficial?
>
> In our experiments, we find that incorporating structural filters is really useful, which is an exploitation mechanism. Our filters were inspired by those implemented in [1], which remove any generated molecules that are extremely dissimilar from the current top molecules. The point of incorporating diversity mechanisms is interesting. Per this suggestion, we implemented our operators in a second genetic algorithm framework, JANUS, which maintains an exploration population. We find that incorporating our operators into JANUS improves its performance, which we show below and in Appendix D3. This indicates that our method works synergistically with a range of evolutionary mechanisms.
>
>  | JANUS \\ Objective (↑) | w/o LLM       | w/ MolSTM       | w/ BioT5         |
> |--------------------------------|---------------|----------------|------------------|
> | JNK3                           | 0.678 ± 0.031 | 0.680 ± 0.024  | **0.685 ± 0.044** |
> | albuterol_similarity           | 0.712 ± 0.049 | 0.779 ± 0.038  | **0.904 ± 0.051** |
>
>
> ### Closing comment
> We are happy to address any additional comments or concerns the reviewer may have. Otherwise, we would appreciate if the reviewer continues to view this work positively. Thank you once again for your time and thoughtful feedback.
>
> ### References
>
> [1] Nigam, AkshatKumar, Robert Pollice, and Alán Aspuru-Guzik. "Parallel tempered genetic algorithm guided by deep neural networks for inverse molecular design." Digital Discovery 1.4 (2022): 390-404.

---

> > ### Author Response · Authors · 2024-11-25
> > **Looking Forward to Your Feedback**
> >
> > Dear Reviewer,
> >
> > We would like to thank you for the time you have spent reviewing our paper so far. As the rebuttal period is ending soon, we would like the opportunity to clarify any remaining questions or concerns. We have incorporated the reviewer’s important suggestion of a time analysis of our method, where we show that as objective function time increases, the cost of LLM inference is negligible. Per your suggestion, we have also shown that our method can improve existing backbones consisting of diverse evolutionary mechanisms.
> >
> > We would be happy to engage in further discussions, please let us know if additional questions or concerns arise! Thank you again for your positive outlook of our paper and we are thrilled if you would continue your enthusiastic support of it.

---

### Official Review · Reviewer_XR1K · 2024-10-31

**Soundness:** 3
**Presentation:** 3
**Contribution:** 3
**Rating:** 6
**Confidence:** 4

**Summary:**

This work discusses an approach to molecular discovery, an optimization problem that faces challenges due to non-differentiable objectives. Evolutionary Algorithms (EAs) typically explore chemical space through random mutations and crossovers, which requires numerous costly evaluations. This work integrates chemistry-aware Large Language Models (LLMs) into EAs, redesigning crossover and mutation operations based on chemical knowledge from LLMs.

**Strengths:**

The strength of the proposed Molecular Language-Enhanced Evolutionary Optimization (MoLLEO) framework lies in its innovative integration of chemistry-aware LLMs (GPT-4, BioT5, MolSTM) into Evolutionary Algorithms (EAs) to improve molecular discovery. By using LLMs as genetic operators in crossover and mutation, MoLLEO enhances proposal quality and accelerates the optimization process. The framework’s effectiveness is demonstrated across multiple black-box optimization tasks, including complex ones like protein-ligand docking, outperforming baseline EAs and other optimization methods.

**Weaknesses:**

1. The diagram lacks clarity. Based on the description, this work employs LLMs designed solely for text, but Figure 1 appears to suggest the use of an MLLM.

2. The presentation of the proposed method should be improved. For instance, there is no justification provided for using different prompts for LLMs like GPT-4, BIOT5, and MOLSTM. Introducing these LLMs could be moved to the appendix, and the comparison of the designed strategies would benefit from a clearer presentation, such as in a table format.

More concerns can be found in the "Questions".

**Questions:**

1. My main concerns is the correctness of this method. How to ensure LLMs can produce better offerspring based on the description and fitness value? I believe the molecular is so complex that it is very difficult to figure out which component contributes to the high fitness.

2. The abstract states that the proposed method addressed problem of expensive evaluation. However, I didn't see any relevatnt discussion on it. It is solved by surrogate model normally in evolutionary computation.

3. I am uncertain about the evaluation function. Could you clarify what is meant by 'black-box'? In my experience, fitness is typically determined either through practical experimentation or by calculation using a specific function.

4. How to turn molecule into text description so that it can be processed by LLMs?

5. I am curious the scale of the molecule and its text description. If the scale is too big, will LLMs suffers the risk of correctness? I mean, the generated molecule is probably infeasible?

---

> ### Author Response · Authors · 2024-11-22
>
> We thank the reviewer for their feedback and thoughtful questions. We are ecstatic that the reviewer finds our method “innovative” and “enhances proposal quality”. We now turn to address the main points raised by the reviewer, with a focus on improving the clarity of our presentation and addressing the question of surrogate models. We have updated our PDF in blue text to highlight these changes as well.
>
>
> > Based on the description, this work employs LLMs designed solely for text, but Figure 1 appears to suggest the use of an MLLM.
>
> We utilize SMILES/SELFIES representations to convert molecules to textual representations for processing by LLMs, which is consistent with how the underlying LLMs trained their models. The molecule graph depicted in the original figure was included solely for visualization; we do not use MLLMs and apologize for the confusion.  Per your suggestion, we have updated Figure 1 to show the conversion from molecule graph to SMILES string before model ingestion.
>
> > The presentation of the proposed method should be improved. For instance, there is no justification provided for using different prompts for LLMs like GPT-4, BIOT5, and MOLSTM.  Introducing these LLMs could be moved to the appendix, and the comparison of the designed strategies would benefit from a clearer presentation, such as in a table format.
>
> We thank the reviewer for providing these suggestions to improve the clarity of our work, although we think it is valuable to describe how each LLM works in the main body since they rely on different inference schemes (i.e., next-token prediction vs. maximizing cosine similarity using gradient ascent). To design the prompts, we followed the best practices proposed in each of the corresponding models and did not deviate from these. For GPT-4, we used similar prompts as in [1], which tested the model in several chemistry settings. We show all prompts used in Appendix E. Per the reviewer’s suggestion, we have mentioned this explicitly in Section 3.2; we hope the reviewer finds this more clear.
>
> > How can we ensure LLMs can produce better offspring based on the description and fitness value?
>
> The reviewer raises an important question – indeed, we cannot guarantee that LLMs produce better offspring based on an objective description. We have documented this in Table A1, where we show that 20-50% of offspring have higher fitness. However, this value is significantly higher than the number of offspring with higher fitness generated by GraphGA (see Figure 2).  We also know that LLMs on their own are not able to generate extremely fit molecules from prompting alone (see Appendix C1, Table A3). However, by incorporating LLMs into an optimization framework,  it’s perfectly fine if some offspring are not better, since they will not proceed through the evolutionary framework. What’s important is that LLMs produce better offspring *with higher likelihood* than by random mutations; these offspring will be propagated through the optimization framework to find better molecules faster.
>
> > The abstract states that the proposed method addressed the problem of expensive evaluation. However, I didn't see any relevant discussion on it. It is solved by surrogate model normally in evolutionary computation.
>
> For the experiments, we use AUC top-10 as the evaluation metric, which is the area under the curve of top-10 average property values versus the number of oracle calls. This metric takes into account both the objective values and the computational budget spent. MolLEO achieves better performance under the same oracle budget, demonstrating improved sample efficiency compared to traditional approaches.
>
> The reviewer is correct in that surrogate models are indeed commonly used in evolutionary frameworks. At the reviewer’s suggestion, we have integrated our method into JANUS [2], which incorporates surrogate models within its framework to filter out crossover offspring by replacing its mutation operator with our LLM-based MolLEO mutation operator on two tasks, which we show in a table below. We find that our method is able to achieve better performance than JANUS on its own, which suggests that our method can be combined with surrogate models to find better molecules within a fixed experimental budget.
>
>
>  | JANUS \\ Objective (↑) | default GA       | JANUS + MolLEO(MolSTM)       | JANUS + MolLEO(BioT5)         |
> |--------------------------------|---------------|----------------|------------------|
> | JNK3                           | 0.678 ± 0.031 | 0.680 ± 0.024  | **0.685 ± 0.044** |
> | albuterol_similarity           | 0.712 ± 0.049 | 0.779 ± 0.038  | **0.904 ± 0.051** |

---

> ### Author Response · Authors · 2024-11-22
>
> > I am uncertain about the evaluation function. Could you clarify what is meant by 'black-box'? In my experience, fitness is typically determined either through practical experimentation or by calculation using a specific function.
>
> Certainly, we are happy to provide this clarification. By "black-box," we mean that the function's closed-form expression is unknown, and its internal workings are not directly accessible. In practical scenarios, fitness is typically obtained through experimentation (e.g., a bioassay) or simulation (e.g. DFT). As we cannot assume the availability of a closed-form function or its gradients, our approach treats it as a black-box optimization problem. In our study, we use simulations (protein-ligand docking) and simpler computational models, where we assume we don’t have access to gradients.
>
> > How to turn molecule into text description so that it can be processed by LLMs?
>
> The transformation of molecular structures into text-based representations is commonly achieved using the Simplified Molecular Input Line Entry System (SMILES) notation [3]. SMILES provides a compact string representation of 2D chemical structures that is both human-readable and amenable to computational processing. LLMs trained on datasets containing SMILES notations learn the joint probability distribution of these molecular representations alongside corresponding text descriptions of their properties and functions. This allows LLMs to encode both molecular structure and textual information together.
>
> > I am curious the scale of the molecule and its text description. If the scale is too big, will LLMs suffers the risk of correctness? I mean, the generated molecule is probably infeasible?
>
> The reviewer's concern about the correctness of molecules generated by LLMs is very reasonable, and is a challenge with generative models in general. As noted in Appendix C.1, we cannot guarantee that an LLM generates a valid molecule with a single query. To address this, we fall back to random mutations if the LLM-generated molecule is invalid; we also experiment with using chemical languages that always guarantee generating valid molecules (SELFIES, which is used in BioT5). This LLM generates the strongest results from the open-source models.
>
> Regarding the feasibility of generated molecules, it is important to note that this is a broader challenge inherent to all generative models, irrespective of molecule size or length. While synthesizability is a recognized issue in molecular generation, it falls outside the primary scope of this work.
>
> ### Closing comment
> We sincerely thank the reviewer for their valuable feedback. We hope our rebuttal has effectively addressed their questions and concerns. If the reviewer is satisfied with our responses, we kindly ask them to consider a reassessment of our paper. We remain more than happy to address any additional questions or concerns that may arise.
>
>
> ### References
>
> [1] Microsoft AI4Science et al. The impact of large language models on scientific discovery: a preliminary study using gpt-4. arXiv, 2023.
>
> [2] Nigam, AkshatKumar, Robert Pollice, and Alán Aspuru-Guzik. "Parallel tempered genetic algorithm guided by deep neural networks for inverse molecular design." Digital Discovery 1.4 (2022): 390-404.
>
> [3] Weininger, David. "SMILES, a chemical language and information system. 1. Introduction to methodology and encoding rules." Journal of chemical information and computer sciences 28.1 (1988): 31-36.

---

> > ### Author Response · Authors · 2024-11-25
> > **Looking Forward to Your Feedback**
> >
> > Dear Reviewer,
> >
> > We are very appreciative of the time you have spent evaluating our work so far. The rebuttal period is ending quite soon, and so we wanted to reach out and see if the reviewer had any remaining questions or concerns. We would like to note that in our rebuttal, we have taken into account the reviewer's suggestions of clarifying how we present important aspects of our method (transforming molecules into language, what happens if good offspring are not produced), as well as discussing the efficacy of surrogate models. With your suggestion, we have also run our method on a backbone that incorporates surrogate models and find that it improves performance.
> >
> > If the reviewer has any remaining doubts or requires further clarification regarding these updates, we would be happy to engage in further discussions. We thank the reviewer again for spending time reviewing our paper. If the reviewer finds our updates to be satisfactory in addressing their concerns, we would be very appreciative if the reviewer would consider raising their score.

---

> > ### Comment · Reviewer_XR1K · 2024-11-26
> >
> > Thank you for your response. I will raise my score.

---

> > > ### Author Response · Authors · 2024-11-26
> > >
> > > Thank you for your positive feedback! We truly appreciate your support and time in reviewing our work.

---

### Official Review · Reviewer_wNKW · 2024-11-01

**Soundness:** 2
**Presentation:** 2
**Contribution:** 2
**Rating:** 6
**Confidence:** 4

**Summary:**

This paper introduces MOLLEO (Molecular Language-Enhanced Evolutionary Optimization), which is a framework that combines Large Language Models (LLMs) with evolutionary algorithms for molecular discovery. The study integrates three LLMs - GPT-4, BioT5, and MoleculeSTM - into genetic operators such as mutations and crossover operations for molecular generation. The experimental results span multiple tasks, including property optimization, molecular rediscovery, and structure-based drug design, where MOLLEO demonstrates superior performance over its baseline models. Among the tested models, MOLLEO with GPT-4 achieves the best results in 15 out of 23 tasks. Furthermore, this study tackles more challenging scenarios such as protein-ligand docking tasks, which better represent real-world molecular generation conditions.

**Strengths:**

The primary strength of this paper is its straightforward implementation of integrating LLMs into evolutionary algorithms for molecular optimization. This makes it easy to follow and understand the core mechanism of the proposed framework. Another strength is the framework's flexibility in incorporating different types of LLMs, from commercial (GPT-4) to open-source models (BioT5 and MoleculeSTM), making it adaptable to different resource constraints and use cases. This study also demonstrates practical applicability by showing how MOLLEO can improve upon existing molecules in existing chemical datasets like ZINC 250K, indicating its potential value for real-world drug discovery pipeline.

**Weaknesses:**

The weakness of this paper is particularly regarding the novelty of the methodology. The core idea of MOLLEO is essentially a straightforward substitution, where genetic operators in the existing evolutionary algorithm framework (Graph-GA) are replaced with LLMs. Fundamentally, this method mainly involves leveraging LLMs to perform mutation and crossover operations, which could be seen as a relatively simple extension of existing approaches.

Furthermore, the paper lacks a detailed explanation of how the authors address multi-objective optimization settings. While they acknowledge that determining the weight for each objective function in a sum-aggregate objective could be challenging, they do not specify how these weights are assigned in their experiments.

Additionally, the authors propose handling multi-objective settings using Pareto set selection, where only the Pareto frontier of the current population is retained. However, this approach is a standard practice in evolutionary algorithms, as seen in well-established methods like NSGA [1] and MOEA/D [2], which leverage Pareto ranking. The authors might assume that Pareto set selection is less frequently applied in molecular optimization, this concept has already been widely applied in molecular optimization problems in previous studies [3-5]. Though the authors do not claim a novel contribution in using Pareto set selection, the paper would benefit from a more thorough discussion of their approach to multi-objective optimization and clearer positioning within the broader landscape of Pareto-based optimization approaches.

[1] Deb, Kalyanmoy, et al. "A fast and elitist multiobjective genetic algorithm: NSGA-II." IEEE transactions on evolutionary computation 6.2 (2002): 182-197.

[2] Zhang, Qingfu, and Hui Li. "MOEA/D: A multiobjective evolutionary algorithm based on decomposition." IEEE Transactions on evolutionary computation 11.6 (2007): 712-731.

[3] Sun, Mengying, et al. "Molsearch: search-based multi-objective molecular generation and property optimization." Proceedings of the 28th ACM SIGKDD conference on knowledge discovery and data mining. 2022.

[4] Zhu, Yiheng, et al. "Sample-efficient multi-objective molecular optimization with gflownets." Advances in Neural Information Processing Systems 36 (2023).

[5] Shin, Dong-Hee, et al. "DyMol: Dynamic Many-Objective Molecular Optimization with Objective Decomposition and Progressive Optimization." ICLR 2024 Workshop on Generative and Experimental Perspectives for Biomolecular Design.

**Questions:**

1) The authors propose using LLMs for genetic operators (mutation and crossover), but how does this approach fundamentally differ from existing methods that use deep neural networks for learning genetic operators? What unique capabilities do LLMs bring to molecular generation that cannot be achieved through existing deep learning-based mutation and crossover operations? The authors may consider that LLMs do not require additional training for genetic operator learning, as they can be adapted simply by modifying prompts. However, I believe that the inference process for LLMs could introduce a computational cost comparable to training deep neural networks for genetic operators. Furthermore, while the authors may suggest that LLMs can facilitate the generation of molecules aligned with objective fitness functions through descriptive prompts, deep neural networks trained specifically for genetic operations can offer distinct advantages. By learning from a dataset of molecules, they can capture meaningful molecular representations and generate novel combinations that are data-driven rather than randomly assembled.

2) The authors state that in Graph-GA, the default crossover and mutation operators are pre-defined by domain experts based on chemical knowledge. How can we be confident that LLMs naturally incorporate this domain-specific knowledge and execute these crossover and mutation operations accurately, rather than merely presenting empirical results? What evidence supports the notion that LLMs inherently understand and replicate these chemically-informed operations?

3) The authors state that they examined whether the open-source models generated molecules previously seen during training and found no overlap between generated molecules and the datasets. However, given that LLMs are trained on extensive chemical literature, including popular molecular objectives like JNK3, GSK, QED, and DRD2, could there still be a risk of data leakage? Specifically, if LLMs have learned about these well-known objectives, they may inherently recognize and prioritize molecular structures that are known to perform well for these targets. If the LLM’s prior knowledge is being leveraged intentionally, how does this approach handle genuinely novel molecular objectives that the LLM has never encountered? In cases where the LLM lacks specific training for unfamiliar objectives, how can it be expected to perform effective genetic operations without relying on prior knowledge, especially when such knowledge may be absent or limited?

4) Building on the previous question, methods like REINVENT and Augmented Memory can optimize molecules from scratch, allowing them to adapt effectively to entirely new and unseen objectives. Given that MOLLEO relies on LLMs with pre-existing chemical knowledge, can it perform better when optimizing for novel molecular targets with limited or no prior information? How does MOLLEO address this challenge to ensure effective adaptation and optimization for objectives that the model has never encountered?

5) The authors employ GraphGA as the backbone model, but why was this specific choice made? Other genetic algorithm-based methods, such as GPBO, or even other generative models like Augmented Memory, could also be suitable. For instance, in molecular generation, LLM-based genetic operations could be implemented and integrated into a replay buffer to iteratively refine results. This approach seems feasible, so why did the authors choose to rely solely on GraphGA as the backbone model?

6) In their multi-objective experiments (shown in Table 2), where mixed objectives are used (e.g., maximizing QED while minimizing SAscore), the authors do not explicitly explain how they handle the differences between minimization and maximization objectives. Do they simply flip the sign of the minimization objectives to treat them as maximization objectives? If so, is this approach sufficient for effectively balancing objectives with inherently different goals?

7) In Figure 2, there appears to be a typo where "LLM editting" is used. Should this perhaps be corrected to "LLM editing"?

---

> ### Author Response · Authors · 2024-11-22
>
> We are extremely appreciative of the reviewer’s feedback and suggestions to improve our work. We are glad that the reviewer finds our work “easy to follow” and that our framework is “flexible” for “different resource constraints”. We are also heartened that the reviewer considers our work to have “potential value for real-world drug discovery pipeline”.
>
> Below, we address the reviewer’s concerns about task familiarity and include several additional baselines, at the reviewer’s suggestions. We also highlight all these changes in blue in the updated PDF.
>
> > The core idea of MOLLEO is essentially a straightforward substitution, where genetic operators in the existing evolutionary algorithm framework (Graph-GA) are replaced with LLMs, which could be seen as a relatively simple extension of existing approaches.
>
> We thank the reviewer for this constructive feedback, although we argue that incorporating LLM-based mutations into black-box optimization frameworks has not been done before in scientific discovery to our knowledge. Evolutionary algorithms (EAs) are widely used in chemistry [1], although they traditionally rely on random manipulations of chemical structures, which can be inefficient since chemical space is large and experiments are expensive. LLMs have been trained on large amounts of chemical knowledge and it has previously been shown that they can tweak molecular structures based on text descriptions, and so we wanted to investigate whether or not these tweaks are informative enough to be incorporated into optimization algorithms. Up until now, the incorporation of LLMs in EAs has primarily been shown for code/algorithm design [2, 3], but not in scientific discovery, which we argue is a much harder domain and it was not obvious whether language-guided mutations would be useful. Figuring out how to incorporate them is nontrivial; we did several ablation studies and showed that our model is able to beat many existing methods on an established benchmark and on complicated property optimization tasks, including protein-drug interaction tasks. We hope this answer demonstrates to the reviewer that our work is both novel and useful for the community.
>
> > Authors acknowledge that determining the weight for each objective function in a sum-aggregate objective could be challenging, they do not specify how these weights are assigned in their experiments.
>
> In our experiments, we assigned equal weights to each objective in the sum-aggregate objective function. This was to avoid introducing additional hyperparameters since determining objective weights can be nontrivial.  We instead compute this simple metric, as has been done in other works [4], and focus on more robust multi-objective optimization algorithms such as Pareto set selection.
>
> > [Pareto set selection] is a standard practice in evolutionary algorithms, as seen in well-established methods like NSGA [1] and MOEA/D [2], which leverage Pareto ranking … Though the authors do not claim a novel contribution in using Pareto set selection, the paper would benefit from a more thorough discussion of their approach to multi-objective optimization and clearer positioning within the broader landscape of Pareto-based optimization approaches.
>
> We thank the reviewer for pointing us to these works. We agree that there may be several interesting multiobjective optimization algorithms to explore, although as the reviewer pointed out, this is not a contribution we claim in this work. In this paper, we opt for the most standard multi-objective optimization methods that are widely used in practice. However, the works that the reviewer points out are interesting. To our understanding, PO differs from these by simply keeping all the solutions in the Pareto set, while the other methods incorporate some selection criteria of clustering for selecting the set.  Motivated by this, we have created a new related works section on multiobjective algorithms (Appendix A1) and think they would be interesting to investigate in a separate work.
>
> > What unique capabilities do LLMs bring to molecular generation that cannot be achieved through existing deep learning-based mutation and crossover operations? The authors may consider that LLMs do not require additional training for genetic operator learning, as they can be adapted simply by modifying prompts. However, I believe that the inference process for LLMs could introduce a computational cost comparable to training deep neural networks for genetic operators. Furthermore, while the authors may suggest that LLMs can facilitate the generation of molecules aligned with objective fitness functions through descriptive prompts, deep neural networks trained specifically for genetic operations can offer distinct advantages. By learning from a dataset of molecules, they can capture meaningful molecular representations and generate novel combinations that are data-driven rather than randomly assembled.

---

> ### Author Response · Authors · 2024-11-22
>
> The reviewer asks a very interesting question. Our initial hypothesis was that LLMs themselves are a useful deep-learning based approach for generating mutation and crossover operations because they have been trained on large-scale functional text. The cool thing about this is LLMs can link molecular structures with a description of their functions, and the outcome is that molecules with similar functions are closer together in their latent representations.
>
> The challenge with existing deep-learning based operators is  that they are typically trained on small-scale datasets for very specific tasks, and introducing a new task might mean needing to re-train the network. On the other hand, the breadth of training data in LLMs also offers an advantage; if there is data scarcity for labels in some task, there can be information shared from other tasks that generate meaningful molecular representations.
>
> In our evaluation, we do compare our method to DNN-based approaches (REINVENT, DST) and find that our method outperforms both. If the reviewer has a specific work they are referring to, we would be happy to take a look.
>
> While the reviewer raises a very reasonable point about computational costs, we find that for tasks where the oracle call is expensive, the call time to the LLM becomes negligible. For example, the docking experiments for DRD3 (PDB ID 3pbl) on average took 8 h 23m ± 1 h 46 m on average for 5 runs for the baseline GraphGA, while it tool MolLEO(BIOT5) 6 h 57m ± 1 h 24 m on average. Thus, the bottleneck in those experiments was not the LLM call, but rather the time to run the docking simulation. We hope this quantitative evaluation provides justification for the reviewer as to why the runtime of LLMs is not prohibitive.
>
> > How can we be confident that LLMs naturally incorporate … domain-specific knowledge and execute these crossover and mutation operations accurately, rather than merely presenting empirical results? What evidence supports the notion that LLMs inherently understand and replicate these chemically-informed operations?
>
> We thank the reviewer for raising this valid concern. Our goal is not actually to necessarily replicate these inductive biases. Instead, our motivation is that LLMs are able to provide *functionally-informed* operations because they were trained on large amounts of paired molecule-function data, and so LLMs can learn a meaningful landscape of molecular structure representations that are conditioned on their functions and encode domain-specific knowledge. Thus, when prompted with a new molecule structure, the LLM can edit the molecule in a way that improves its function, either by gradient descent or maximizing token probabilities.
>
> We measure the ability of LLMs to provide good functionally-informed operations through several ablations and experiments (Figure 2, Table A1) by having our operations perform a single round of edits on a set of molecules and measuring the functions of molecule pairs before and after, and find that they outperform pre-defined, domain-expert operations. This may also suggest that it may not actually be necessary to encode these inductive biases, which interestingly is similar in spirit to what was found in the AlphaFold3 paper [5].
>
> > If LLMs have learned about these well-known objectives, they may inherently recognize and prioritize molecular structures that are known to perform well for these targets. If LLMs have learned about … well-known objectives, they may inherently recognize and prioritize molecular structures that are known to perform well for these targets. If the LLM’s prior knowledge is being leveraged intentionally, how does this approach handle genuinely novel molecular objectives that the LLM has never encountered?
>
> We thank the reviewer for these thoughtful and nuanced questions. While LLMs have been trained on extensive chemical literature, they are actually not good at generating top candidate molecules on their own, which we show in App. Table A3. Thus, it is doubtful that they are simply retrieving instances from the training data.
>
> Per the reviewer’s suggestion, we did an analysis for which protein-drug interaction tasks appear in the training datasets of MoleculeSTM and BioT5 in Appendix Table A14. Interestingly, we find that their training datasets do not explicitly contain captions related to two of the targets, DRD3 and Adenosine receptor A2a, although they contain information to related concepts, such as other adenosine receptors and dopamine receptors. This indicates that we can leverage information from related tasks to optimize molecules on new, unseen tasks.

---

> ### Author Response · Authors · 2024-11-22
>
> One key strength of LLMs is their ability to generalize to several target functions. In the event that we are optimizing for a truly novel task that the LLM has never seen before, the assumption is that the LLM could have been trained on other tasks that are related (for example, say we are optimizing for Jun N-kinase 3 but the LLM has only seen other kinases). These related tasks can still encode some useful information about kinases in general, which can allow for meaningful, functionally-informed operations. This is an example of transfer learning. If the task is truly novel (i.e., all tasks that the LLM has been trained on have orthogonal information), then at worst the LLM spits out random molecules, which can be seen as random mutation operations. This is also the benefit of plugging LLMs into an optimization framework – edits that are not meaningful are filtered out.
>
> We also test going beyond what chemical information is currently known by initializing our optimization framework with the best molecules currently known that are JNK3 inhibitors in Table 3 of the main paper, a situation that most resembles a new discovery setting. We find that our method is able to find better new molecules than the baseline.
>
> > The authors employ GraphGA as the backbone model, but why was this specific choice made? Other genetic algorithm-based methods, such as GPBO, or even other generative models like Augmented Memory, could also be suitable.
>
> We selected GraphGA as the backbone model because its simplicity as a baseline allows us to focus on and directly evaluate the effects of the LLM-based genetic operators in our study. Our focus in this work was on understanding the impact of different LLM-based genetic operators within a simple framework.
>
> As the reviewer has rightly suggested, it would be interesting to explore our mutations in other settings. To this end, we have incorporated our LLM-based genetic operators in three additional backbones: Augmented Memory [6], Genetic GFN [7] (as suggested by Reviewer bF3X), and JANUS [8] (which incorporates exploration mechanisms, as suggested by Reviewer TwFV).  The results are shown below; we also added them to Appendices D2 and D3 in the updated manuscript.
>
>  | Augmented Memory \\ Objective (↑) | w/o LLM       | w/ BioT5       | w/ GPT-4         |
> |--------------------------------|---------------|----------------|------------------|
> | JNK3                           | 0.773 ± 0.073 | 0.781 ± 0.094  | **0.794 ± 0.087** |
> | albuterol_similarity           | 0.918 ± 0.026 | 0.925 ± 0.076  | **0.941 ± 0.033** |
>
>
> | Genetic GFN \\ Objective (↑)       | w/o LLM       | w/ BioT5   | w/ GPT-4      |
> |-----------------------------|------------------|-------------------|---------------------|
> | JNK3                        | 0.766 ± 0.077    | 0.775 ± 0.056     | **0.783 ± 0.034**   |
> | Albuterol Similarity        | 0.946 ± 0.013    | 0.962 ± 0.017     | **0.971 ± 0.020**   |
>
>
>  | JANUS \\ Objective (↑) | w/o LLM       | w/ MolSTM       | w/ BioT5         |
> |--------------------------------|---------------|----------------|------------------|
> | JNK3                           | 0.678 ± 0.031 | 0.680 ± 0.024  | **0.685 ± 0.044** |
> | albuterol_similarity           | 0.712 ± 0.049 | 0.779 ± 0.038  | **0.904 ± 0.051** |
>
>
> We hope the reviewer finds these results interesting that LLM-based generation operations are beneficial in several existing frameworks.
>
> >Authors do not explicitly explain how they handle the differences between minimization and maximization objectives. Do they simply flip the sign of the minimization objectives to treat them as maximization objectives? If so, is this approach sufficient for effectively balancing objectives with inherently different goals?
>
> We confirm that we do indeed flip the signs of the minimization objectives to treat them as maximization objectives. While this approach is a straightforward transformation, it is a setting that we compare against other methods to treat objectives with the same weight, which is a reasonable assumption with no additional prior information (i.e., determining different weights for different objectives is nontrivial).
>
> > In Figure 2, there appears to be a typo where "LLM editting" is used. Should this perhaps be corrected to "LLM editing"?
>
> We thank the reviewer for pointing out the typo in Figure 2; we have corrected it.
>
> ### Closing comments
>
> We thank the reviewer for the extremely valuable feedback. We hope our rebuttal sufficiently addresses their questions and concerns, and we kindly ask the reviewer to consider a revised evaluation of our paper if the reviewer is satisfied with our responses. We are happy to address any further questions or points of clarification.

---

> ### Author Response · Authors · 2024-11-22
>
> [1] Hiener, Danielle C., and Geoffrey R. Hutchison. "Pareto optimization of oligomer polarizability and dipole moment using a genetic algorithm." The Journal of Physical Chemistry A 126.17 (2022): 2750-2760.
>
> [2] Romera-Paredes B, Barekatain M, Novikov A, et al. Mathematical discoveries from program search with large language models[J]. Nature, 2024, 625(7995): 468-475.
>
> [3] Fernando C, Banarse D, Michalewski H, et al. Promptbreeder: Self-referential self-improvement via prompt evolution[J]. arXiv preprint arXiv:2309.16797, 2023.
>
> [4] Nigam, AkshatKumar, Robert Pollice, and Alán Aspuru-Guzik. "Parallel tempered genetic algorithm guided by deep neural networks for inverse molecular design." Digital Discovery 1.4 (2022): 390-404.
>
> [5] Abramson, Josh, et al. "Accurate structure prediction of biomolecular interactions with AlphaFold 3." Nature (2024): 1-3.
>
> [6] Guo, Jeff, and Philippe Schwaller. "Augmented Memory: Sample-Efficient Generative Molecular Design with Reinforcement Learning." Jacs Au (2024).
>
> [7] Kim, Hyeonah, et al. "Genetic-guided GFlowNets: Advancing in Practical Molecular Optimization Benchmark." arXiv preprint arXiv:2402.05961 (2024).
>
> [8] Nigam, AkshatKumar, Robert Pollice, and Alán Aspuru-Guzik. "Parallel tempered genetic algorithm guided by deep neural networks for inverse molecular design." Digital Discovery 1.4 (2022): 390-404.

---

> > ### Author Response · Authors · 2024-11-25
> > **Looking Forward to Your Feedback**
> >
> > Dear Reviewer,
> >
> > We are really grateful for the time you have spent thoroughly reviewing our paper and providing valuable feedback. As the rebuttal period is quickly approaching the end, we would like to have the opportunity to address any remaining concerns that you have. In our rebuttal, we have implemented your great suggestions of incorporating our method in three additional backbone frameworks (and find that our method improves upon them) and analyzed the generalization capabilities of our framework in terms of task coverage in the training set, demonstrating that it is doing well on beyond tasks that were explicitly present in training. We have also analyzed the time cost of our method, showing that with more expensive objectives, the time for LLM inference is negligible, as well as improved our discussion on multiobjective frameworks.
> >
> > We would be happy to engage in further discussions on these topics or any other important points–please let us know! We thank the reviewer for time spent providing feedback on our paper. If the reviewer finds that our updates are sufficient in answering their questions and addressing concerns, we would really appreciate it if the reviewer could potentially consider a revised evaluation of our paper.

---

> > > ### Comment · Reviewer_wNKW · 2024-11-26
> > >
> > > Thank you for conducting such extensive additional experiments; I understand it must have been challenging, and I sincerely appreciate your effort. However, I have one additional question for further clarification.
> > >
> > > I may have overlooked this detail in the manuscript, but I am curious about the role of mutation in overall performance. Specifically, does mutation contribute significantly to performance, or is its benefit relatively marginal? Could crossover alone be sufficiently effective? This question arises because it seems that the impact of mutation is somewhat influenced by hyperparameters, such as the number of top-scoring candidates selected for mutation.
> > >
> > > While I am not suggesting that incorporating mutation is detrimental, if crossover alone provides substantial benefits, would it be a reasonable approach to focus exclusively on crossover and omit mutation? This could potentially simplify the workload for LLM, allowing it to concentrate solely on the crossover operator. To be clear, I am not questioning why mutation was used even if its contribution might be marginal; rather, I am genuinely curious about whether focusing solely on crossover could still achieve comparable results.
> > >
> > > Additionally, I reviewed the Extended Related Work section in the appendix and appreciate the authors' effort in contextualizing their work within Multi-objective Optimization frameworks. However, the authors state:
> > >
> > > ```
> > > Zhu et al. (2024) introduced a method to decompose the optimization process into a progressive sequence based on the order of objectives. Shin et al. integrated GFlowNets with a preference-conditioned sum of objective functions, further advancing the optimization landscape.
> > > ```
> > > However, I believe there might be an error in attribution. Based on my understanding, it seems that Zhu et al. [1] integrated GFlowNets with a preference-conditioned sum of objective functions.
> > >
> > >
> > > [1] Yiheng Zhu, Jialu Wu, Chaowen Hu, Jiahuan Yan, Tingjun Hou, Jian Wu, et al. Sample-efficient multi-objective molecular optimization with gflownets. Advances in Neural Information Processing Systems, 36, 2024.

---

> > > > ### Author Response · Authors · 2024-11-26
> > > >
> > > > The reviewer asks an excellent question in terms of operator impact, and we agree that studying the role of each operator is very interesting.
> > > >
> > > > In Table A2, we show an ablation of this where we turn on and off different LLM operators. The experiment that we believe the reviewer is suggesting is the second row of that table, where we only apply LLM crossovers and a random mutation with probability=0.067 (this random mutation and probability is from the original Graph GA paper and we keep it because we found that it is helpful for preventing crossover operations from generating offspring that get stuck in local minima). This row demonstrates that for open-source LLMs (BioT5, MolSTM), incorporating LLM crossovers is not as good as incorporating LLM mutations, which we show in the second last row of Table A2 (we see a drop in performance in 3/4 tasks). We suspect that this is because these LLMs were only trained on single molecule objectives, and so processing two molecules as input for crossover might be too nonsensical for the models, although we think this would be interesting to investigate in future work. Thus, when incorporating open-source LLMs into MolLEO, we find that mutation operators are indeed more helpful than crossover in our setup, and so that’s what we proceed with for the remaining experiments.  In terms of the filter hyperparameter, we note that previous works have also incorporated similar setups where the number of molecules to pass through a filter is treated as a hyperparameter [1]. For GPT-4, however, there was no drop in performance when doing crossover; we found it did well with both crossover and/or mutations (possibly due to its ability of processing larger amounts of text in its input), and so we went with only using crossovers to reduce the number of API calls and empirically it did slightly better.
> > > >
> > > > We are also very grateful to the reviewer for pointing out our error in assigning the correct references–thank you for catching this!  We have corrected these citations in our updated manuscript.
> > > >
> > > > We really appreciate the reviewer’s thoughtful feedback and interesting questions. If there is anything else we can clarify or elaborate on regarding our mutation/crossover ablation (or anything else), please let us know!
> > > >
> > > > [1] Nigam, AkshatKumar, Robert Pollice, and Alán Aspuru-Guzik. "Parallel tempered genetic algorithm guided by deep neural networks for inverse molecular design." Digital Discovery 1.4 (2022): 390-404.

---

> > > > > ### Author Response · Authors · 2024-11-30
> > > > > **Kindly awaiting feedback**
> > > > >
> > > > > Dear Reviewer,
> > > > >
> > > > > We sincerely appreciate the time and effort you have dedicated to providing constructive and valuable feedback on our paper so far. We wanted to check in to see if the reviewer has any remaining questions or concerns since the extended discussion period is ending soon, particularly regarding your recent inquiries about mutations vs crossover operations.
> > > > >
> > > > > Once again, we thank you for your thoughtful evaluation of our work. If you find our new experiments and discussions useful in enhancing the quality of the paper, we would be truly grateful if you might consider a revised evaluation.

---

> > > > > > ### Comment · Reviewer_wNKW · 2024-12-02
> > > > > >
> > > > > > Dear authors, thank you for your responses and I have raised my score to 6 accordingly.

---

> > > > > > > ### Author Response · Authors · 2024-12-03
> > > > > > >
> > > > > > > We thank the reviewer for their feedback and positive endorsement of our work!  We are happy to answer any other questions that may arise, and are very appreciative to the reviewer for their time and effort during this rebuttal period.

---

### Author Response · Authors · 2024-11-22
**General response by Authors**

We sincerely thank all reviewers for their valuable time and constructive feedback. We are thrilled that the reviewers recognized the significance of our work in integrating LLMs into EAs for molecular optimization, with a framework that is both easy to follow and adaptable to diverse constraints and use cases (R wNKW). We are pleased that our innovative integration of chemistry-aware LLMs into evolutionary processes was acknowledged for improving proposal quality and accelerating molecular discovery (R XR1K). Additionally, we greatly appreciate the recognition of MolLEO’s novel application of LLMs within EAs, achieving superior performance across 26 molecular design tasks and demonstrating promise for early drug discovery (R TwFV).

Below, we address shared concerns, summarize new experiments, and highlight corresponding updates in the updated manuscript, which are marked in blue.

- As suggested by Reviewers **wNKW, bF3X, and TwFV, we have included** results on incorporating MolLEO into two other, more recent genetic algorithms – Genetic GFN [1] and JANUS [2] (see Appendix D.3 in the updated manuscript). We find that incorporating our genetic operators improves the performance of the backbone GA.

- As suggested by Reviewers **wNKW, bF3X, we also incorporate** our LLM-based genetic operators into Augmented Memory to refine the molecules in the replay buffer [3] (see Appendix D.2 in the updated manuscript). Again, we are excited to report that incorporating our operators improves model results.

- Reviewer **bF3X provided a fascinating suggestion of analyzing molecules before and after editing to determine how different they are**. We report **average Tanimoto similarities between molecules before and after editing vs. with random molecules** and find that edited molecules are closer to parent molecules than they are to random molecules, indicating that edits preserve structural elements. We show these results in Appendix D.5.

- **Reviewers TwFV and wNKW brought up an important question about computational cost. We analyzed experiment times and found that as experiments become more expensive (i.e., docking experiments)**,  the cost of the LLM call becomes insignificant in comparison to the docking time, hence the runtime is governed by the oracle call. We report the exact times in Appendix D.4.

- As suggested by Reviewers **wNKW and bF3X, we analyzed which tasks have captions that appear in the training set**. We found that **even for tasks with no captions in the training set,  our method still performs better than the underlying GA**, indicating that **there is potentially useful information being shared from related tasks and underscores the utility of our model in novel settings** (see Appendix D.4 in the updated manuscript).

We hope that our additional experiments and descriptions address any remaining reviewer concerns and we look forward to further discussing with them. We would like to thank all reviewers for their valuable time and effort in reviewing our manuscript!

[1] Kim, Hyeonah, et al. "Genetic-guided GFlowNets: Advancing in Practical Molecular Optimization Benchmark." arXiv preprint arXiv:2402.05961 (2024).

[2] Nigam, AkshatKumar, Robert Pollice, and Alán Aspuru-Guzik. "Parallel tempered genetic algorithm guided by deep neural networks for inverse molecular design." Digital Discovery 1.4 (2022): 390-404.

[3] Guo, Jeff, and Philippe Schwaller. "Augmented Memory: Sample-Efficient Generative Molecular Design with Reinforcement Learning." Jacs Au (2024).

---

### Meta-Review · Area_Chair_v4pN · 2024-12-25

**Metareview:**

The paper combines genetic algorithms (GraphGAN) with LLMs for molecular optimization , where  crossover and mutations are replaced by LLMs, yielding an improvement in sample efficiency of evolutionary search. The idea is simple and over the rebuttal the methodology was improved and ablated using reviewers suggestions (using different genetic algorithms, augmented memory, computing distance before and after optimization of the molecules wwrt to a random molecules baselines, reporting computational costs ).

All reviewers agreed that this is a good paper and that it is a nice and novel applications of LLM proposals within evolutionary algorithms.

**Additional Comments On Reviewer Discussion:**

reviewers increased their scores in the rebuttal that answered their questions and authors provided substantial improvements and experimenantation.

---

### Decision · Program_Chairs · 2025-01-22

Accept (Poster)